# Mitigating Biases in Blackbox Feature Extractors for Image Classification Tasks

Abhipsa Basu     Saswat Subhajyoti Mallick *    R. Venkatesh Babu

Vision and AI Lab, Indian Institute of Science, Bangalore

## Abstract

In image classification, it is common to utilize a pretrained model to extract meaningful features of the input images, and then to train a classifier on top of it to make predictions for any downstream task. Trained on enormous amounts of data, these models have been shown to contain harmful biases which can hurt their performance when adapted for a downstream classification task. Further, very often they may be blackbox, either due to scale, or because of unavailability of model weights or architecture. Thus, during a downstream task, we cannot debias such models by updating the weights of the feature encoder, as only the classifier can be finetuned. In this regard, we investigate the suitability of some existing debiasing techniques and thereby motivate the need for more focused research towards this problem setting. Furthermore, we propose a simple method consisting of a clustering-based adaptive margin loss with a blackbox feature encoder, with no knowledge of the bias attribute. Our experiments demonstrate the effectiveness of our method across multiple benchmarks. The code is publicly available at `https://github.com/abhipsabasu/blackbox_bias_mitigation`.

## 1   Introduction

Deep learning models are to known to inherit harmful stereotypical biases with respect to the different genders, races, cultures, from the datasets they are trained on [1, 2, 3]. For example, a model trained on a gender-biased dataset with images of people having blond and non-blond hair may wrongly learn a correlation between the label 'blond hair' and the gender of the person in the image. Thus the model fails to classify images belonging to minority groups (in this case blond males and non-blond females). These biases can affect the performance of AI systems handling job recruitment, e-commerce, health care, face detection and recognition [4, 5, 6]. Models can also learn spurious correlations between irrelevant training features and the target labels, instead of focusing on the relevant ones [7, 8]. Several works focus on mitigating such biases from trained models in a variety of tasks [9, 10, 11].

In recent times, large-scale models, pretrained on enormous amounts of data, are being used by machine learning practitioners as feature encoders for numerous downstream applications, as their features are shown to be semantically rich [12, 13]. *However, do these powerful pretrained features themselves exude the harmful stereotypical biases that are known to affect traditional deep learning systems?* A previous work [14] sheds light on these questions (when the pretrained model can be fully finetuned on the downstream dataset) – the downstream models finetuned on top of pretrained models can inherit their biases and such biases can be mitigated simply by manipulating the finetuning data.

In this paper, we investigate a more constrained yet practical problem setting. With billion-parameter models gaining impetus in today's world, it may not be feasible to finetune/retrain such models due to scarcity of resources or unavailability of model weights for privacy concerns [13, 12, 15, 16, 17, 18].

---

*Work done while working as a project assistant at the Indian Institute of Science

38th Conference on Neural Information Processing Systems (NeurIPS 2024).

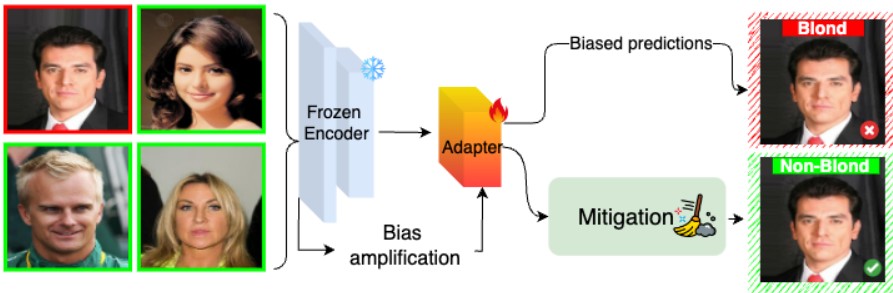

Figure 1: **Summary of our setup**. A frozen pretrained encoder is used as a feature extractor for a downstream task. An adapter is attached on top of the encoder that learns the bias in the downstream dataset. Finally, the bias is mitigated with the help of an adaptive margin loss, leading to unbiased predictions. No knowledge of the bias attribute is assumed apriori.

Hence, for downstream classification tasks, simply the classifier layer on top of the pretrained features is trained on the corresponding dataset, keeping the rest of the network frozen. We ask, *to what extent can such systems be debiased, given that the pretrained model weights are unavailable?*

For the scope of this paper, we assume that the downstream dataset consists of multiple groups (e.g. blond males, non-blond males, blond females, non-blond females) due to the presence of a certain bias attribute (e.g. gender). We find that the effect of the bias in the downstream model depends on how well the pretrained feature encodes the target attribute as compared to the bias attribute. The system remains unbiased if the pretrained features are highly aware of the target attribute (even when the bias correlation in the downstream dataset is high). However, if the pretrained features instead predominantly encode the bias attribute, the downstream system becomes biased. A simple solution to the problem is to group-balance, or reduce the bias-correlation of the finetuning dataset, similar to the suggestion of Wang et al. [14]. However, as manipulating the bias correlation of a dataset is not straightforward, we advocate for a specific strategy to debias the system in the given scenario.

To simplify the bias-mitigation task, we insert a trainable adapter module between the pretrained feature extractor and the classifier. With such a setting, after evaluating existing methods, we find that a large number of these methods do not yield the expected performance gains compared to the ERM-trained model, indicating the challenges in the problem setting as well as the necessity of designing specific debiasing strategies. To aid the mitigation process, we forcefully amplify the bias in the downstream model into the adapter (following previous works [19, 20]), and then investigate a number of simple debiasing techniques utilizing this biased adapter. Finally we propose a method involving clustering-based adaptive margin loss which first clusters the biased feature space, and accordingly sets the margin value for a given sample such that it is inversely proportional to the frequency of the sample's ground truth class in the cluster it belongs to. The problem setting is depicted in Fig. 1. We summarize our key contributions in this paper below:

- We highlight a practical yet under-explored problem setting on how harmful biases creep into a model when it uses pretrained but blackbox feature extractors to obtain features.
- We explore the scenarios in which biases can propagate from the pretrained features to the downstream tasks and demonstrate the necessity and feasibility of debiasing strategies in such cases.
- To debias, we first amplify the bias learnt by the model from the downstream data and propose a simple mitigation strategy by utilizing the identified biases and applying an adaptive margin loss. Extensive experiments show that the proposed method is effective across multiple benchmarks.

## 2 Related Works

**Known biases**: Mitigation of biases with the knowledge of the bias attributes, as well as their labels, is a widely explored problem [21, 22, 23, 24, 25, 26, 27, 28]. Roh et al. [29] propose a sampling algorithm by formulating a combinatorial optimization problem for unbiased sample selection. While Sagawa et al. [23] minimize worst-case training loss over the groups formed using the available bias labels and the target class labels, Zhang et al. [30] disentangle the bias and target representations and debias the network using a mutual information estimator. Some methods adopt a semi-supervised

approach, where bias attributes are annotated for only a few data samples [20, 31]. Another body of work assumes knowledge of the bias attribute but not its specific labels [32, 33, 34, 35].

**Unknown biases**: Recent works consider the more practical scenario of unavailability of the knowledge of the bias attribute as well as its labels [36, 37, 38, 39, 40, 19, 41, 42, 43]. This area encompasses a wide variety of works, a subset of which is described here. JTT [19] debiases models using misclassifications from an ERM-trained model. Some approaches like BPA and GEORGE employ clustering to identify the biases in the dataset [44, 45]. Certain works employ the Generalized Cross-Entropy (GCE) loss [46, 47] to amplify the biases learnt by the network and then debias it. While most of these works, like LfF [20, 37, 48], employ two branches in the network, with one branch over-learning the bias and the other debiasing it, Ahn et al. [40] utilize the GCE loss to learn the bias, and then find the per-sample gradient of the trained model to obtain a balanced dataset. DebiAN [38] identifies multiple biases by having a discoverer model which optimizes an equal opportunity violation loss. Correct-n-Contrast [49] identifies training samples having the same class labels but dissimilar bias features via Empirical Risk Minimization (ERM) training and then applies contrastive loss–similar to Contrastive Adapter [50]–to bring the target features of these same-class samples closer. Qraitem et al [51] propose a sampling method to reduce dataset biases. Kim et al. [52] use a *committee* of auxiliary classifiers to identify the biases in the network, assigning large weights to the identified bias-conflicting samples during the training of the main classifier. Jeon et al. [53] show how bias in a CNN network is more pronounced at the top layers, and hence, leveraging features from the lower layers can help the model to exploit less biased representations. Kirichenko et al. [54] observe that retraining only the last layer is enough for having an unbiased model. However, they require bias annotations in the $2^{nd}$ stage of training. This is avoided by LaBonte et al. [55] by constructing a reweighting dataset using model disagreements for the second stage of training.

**Biases in pretrained models**: Utilizing pretrained feature extractors for downsteam tasks is a common trend in recent times. Such encoders may carry biases that crept in from the datasets used for training them – many recent works focus on quantifying the fairness of such pretrained models [56]. Srinivasan et al. [57] study biases in multimodal vision-language systems. Goyal et al. [58] shows how pretrained models trained with self-supervision exhibit lesser biases than those trained with supervision. Recent works show that biases of pretrained models can be reduced by manipulating the fine-tuning dataset in both Computer Vision and NLP [14, 59]. Salman et al. [60] show that bias transfer happens from pre-trained models to the downstream tasks, even when the target data is unbiased. Our problem setting on the other hand consumes features from a blackbox pretrained encoder and aims to mitigate the biases arising from that encoder. This is challenging, especially when the bias annotations are not available.

## 3 Problem Statement and Methodology

### 3.1 Preliminaries and Problem Setting

This work focuses on the task of image classification. Let $\mathcal{X} = \{x_1, x_2, \ldots x_N\}$ be a set of training images of size $N$, and $\{y_1, y_2, \ldots, y_N\}$ be the corresponding labels, where each $y_i \in \mathcal{C}$ . Each data point $(x_i, y_i)$ is associated with a hidden spurious attribute $a_i \in \mathcal{A}$, and consequently a group $g_i \in \mathcal{G}$, where $\mathcal{G} = \mathcal{C} \times \mathcal{A}$. We assume that *each group $g \in \mathcal{G}$ is present in the training data*. In an unbiased dataset, number of training samples in each $g$ remains approximately equal. However, models tend to learn spurious correlations when there is an imbalance in these numbers. We refer to samples favoured most by the ERM trained models as bias-aligned, and the rest as bias-conflicting [20, 38]. The goal is to train a model to optimize a mapping function $f : \mathcal{X} \to \mathbb{R}^{|\mathcal{C}|}$. In ERM training, we optimize the Cross-Entropy (CE) loss as defined below for a sample $(x, y)$:

$$L_T = \sum_{j=1}^{|\mathcal{C}|} -p_j \log \hat{p}_j \tag{1}$$

where $[p_1, p_2, \ldots, p_{|\mathcal{C}|}]$ is a one-hot vector representing $y$. The corresponding predicted probability vector for the same sample is given by $[\hat{p_1}, \hat{p_1}, \ldots, \hat{p_{|\mathcal{C}|}}]$.

### 3.2 Biases in Pretrained Features

Extracting features from popular pretrained encoders for downstream applications is a common norm, as being trained on large amounts of data equips these models to perform well on a variety of

Table 1: **Performance of Waterbirds on different pretrained encoders**. We compare the model performance for a pretrained ViT-H encoder and a pretrained ResNet-18 encoder for three versions of the training data: a) The original, b) By removing all bias-conflicting (Bi-Co) samples and c) Group-Balanced. For all cases, the test set remains the same.

| Dataset Type | ViT-H | | ResNet-18 | |
|---|---|---|---|---|
| | Worst Group | Average Group | Worst Group | Average Group |
| Original | 88.01 | 94.79 | 38.90 | 76.22 |
| No Bi-Co samples | 80.06 | 90.44 | 18.22 | 68.25 |
| Balanced | 92.37 | 95.47 | 83.55 | 85.94 |

tasks. We freeze these models to avoid backpropagating into their architectures that are generally large-scale [12] and often only accessible through API calls [15, 18, 16, 17]. In this subsection, we choose the WaterBirds dataset [23] to analyse the effect of the pretrained model on the downstream performance (see Section 4 for details). We observe two different scenarios here:

**Scenario 1: Pretrained features are target-aware.** Mehta et al. [61] find that if one chooses a proper pretrained encoder for their specific downstream task, ERM training on a linear classifier attached to the pretrained embedding is enough to obtain unbiased predictions. For instance, using a ViT-H 14 encoder pretrained on the SWAG dataset [62] followed by end-to-end ImageNet [63] finetuning can achieve state-of-the-art results on the Waterbirds dataset. We find that even if all the bias-conflicting samples are removed from the training data, the worst group accuracies remain sufficiently high for the test set (see Table 1). Thus, the pretrained model is highly aware of the downstream target attribute, and no debiasing is required. However, a single pretrained model is hardly a panacea for bias mitigation. For instance, the worst group accuracy for the *CelebA [64] dataset's Blond Hair Classification* is merely $6.71\%$!

**Scenario 2: Pretrained features are bias-aware**. Contrary to *Scenario 1*, when we use an ImageNet-pretrained ResNet-18 as the pretrained encoder, we notice that the worst group accuracies are considerably low for the original Waterbirds dataset itself (see Table 1). An even further drop is seen when the bias-conflicting samples are removed. This shows that this feature encoder exhibits the biases present in the downstream dataset. One simple mitigation strategy is to reduce the bias-correlation in the training set. Group-balancing the Waterbirds dataset leads to considerably uniform accuracies across all groups, irrespective of the underlying pretrained model, as shown in Table 1. This may not always be simple – firstly, the bias attribute may not be known apriori to a practitioner, and secondly, group-balancing or reducing the bias in the downstream dataset may be expensive. Further analyses on these lines are presented in Appendix subsection A.1. Thus, for such systems, an explicit mitigation strategy is required so that the model predictions are unbiased.

### 3.3 Bias mitigation

To mitigate biases in this problem setting, we consider an image classification model that consists of 3 primary components (see Fig. 1): a) A pretrained feature extractor $m$ to extract the image features $f^{\text{locked}} = m(x)$ ($m$ is *blackbox*, i.e. frozen) , b) an adapter consisting of a multi-layer perceptron model $h$ comprising of a single non-linear hidden layer, projecting $f^{\text{locked}}$ to a new latent space defined as $\hat{f} = h(f^{\text{locked}})$, c) a classifier $C$, attached to $\hat{f}$ to obtain the final predictions $\hat{y} = C(\hat{f})$.

**Performance of existing methods**. A large body of work exists in the domain of bias mitigation. However, most of these works assume a fully trainable feature encoder. We pick a few representative ones like DebiaN [38], BPA [45], GEORGE [44], LfF [20], JTT [19] and Contrastive Adapter (Co-Ada) [50] (see Section 2 for details) and explore their efficacy in this setup. We choose three popular benchmarks: Waterbirds, CelebA, and ColorMNIST-0.995 (descriptions available in Sec 4) and use a frozen ResNet-18 feature encoder pretrained on the ImageNet dataset for this experiment. We have a number of key observations (see Table 2):

- LfF performs well on Waterbirds and ColorMNIST-0.995, however, it drops by $14\%$ on CelebA. Similar trends are seen for other methods as well. This shows that existing methods exhibit inconsistencies in this constrained problem setting.

- Co-Ada, which was designed to improve the zero-shot performance of foundation models (without backpropagating into the underlying model), works consistently well across all datasets. However, Contrastive Adapter is computationally expensive (Appendix Table 17).

Table 2: **Performance of existing methods on the proposed problem setting.** We observe that for three different benchmarks, performance of existing methods is either close to that of the ERM model (measured by $\Delta_{\text{ERM}}$), or not consistently high. Co-Ada [50] is one exception among compared methods, having the highest worst-group accuracies for all three benchmarks.

| Backbone | Model | Waterbirds | | CelebA | | CMNIST-0.995 | |
| | | Worst | $\Delta_{\text{ERM}}$ | Worst | $\Delta_{\text{ERM}}$ | Bi-Co | $\Delta_{\text{ERM}}$ |
|---|---|---|---|---|---|---|---|
| ResNet-18 | ERM | 38.90 | 0 | 27.20 | 0 | 49.09 | 0 |
| | DebiAN [38] | 58.94 | 20.04 | 26.10 | −1.1 | 49.82 | 0.73 |
| | BPA [45] | 58.70 | 19.80 | 66.71 | 39.51 | 47.84 | −1.25 |
| | LfF [20] | 66.09 | 27.19 | 13.26 | −13.94 | 71.61 | 22.52 |
| | JTT [19] | 49.84 | 10.94 | 56.25 | 29.05 | 42.86 | −6.23 |
| | GEORGE [44] | 59.35 | 20.45 | 42.22 | 15.02 | 48.77 | −0.32 |
| | Co-Ada [50] | 67.57 | 28.67 | 78.37 | 51.17 | 65.48 | 16.39 |

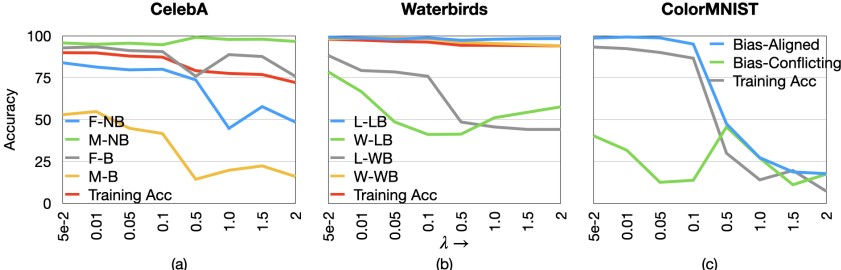

Figure 2: **Effect of increasing weight decay. (a)** For CelebA, as $\lambda$ increases, the worst group (Blond Male (M-B)) accuracy reduces, though the accuracies for Blond Female (F-B) and Non-blond Males (M-NB) still remain high. **(b)** For Waterbirds, we see a fall in both Land-Waterbird (L-WB) and Water-Landbird (W-LB) accuracies with increasing $\lambda$, though the scores for Land-Landbird (L-LB) and Water-Waterbird (W-WB) remain high. **(c)** For ColorMNIST, while the bias-conflicting accuracies reduce as expected, the training accuracy dips beyond $\lambda = 0.1$.

The above observations serve as motivating principles behind the proposed method for this problem setting that is computationally efficient and effective across benchmark datasets.

### 3.4 Our approach

In this subsection, we present our approach. Inspired by previous works [20, 19, 38], we first amplify the biases in the system. Following this, we discuss a few alternatives for mitigation that eventually lead upto our final method.

**Bias-amplified Training**. We propose learning features $\hat{f}$ first using an adapter (i.e. an MLP layer) on top of the pretrained features $f^{\text{locked}}$ through ERM training, and then identifying the biases from these learnt features. The goal is to reduce the worst-group accuracies of the training set while maintaining the overall training performance – thereby amplifying the bias learnt by the model. As indicated in previous works [65, 23], we find that increasing the weight decay $\lambda$ reduces worst-group performance of the training set considerably. We next show the effect of changing $\lambda$ on 3 popular datasets: Waterbirds, CelebA and ColorMNIST-0.995. In Fig. 2(a), we observe that with increasing weight decay, the worst group accuracies reduce for CelebA's Blond Hair classification [64] (i.e. for Blond Males), and also dips for Non-blond Females, while the other two groups' performances remain high. This points to the amplification of gender bias, i.e., the model tends to predict Blond Hair for female images and Non-blond Hair for male images. A similar phenomenon is noticed in the case of Waterbirds [23], as shown in Fig. 2(b). Although the same pattern is exhibited in the case of ColorMNIST-0.995 in Fig. 2(c), an interesting observation also emerges. For $\lambda = 0.1$, training accuracy drops off quite steeply. So, with the goal of reducing the performance on bias conflicting samples, the model may end up not learning anything meaningful at all (as also seen in Fig. 2(a)). This hints at a tradeoff, where we fix $\lambda$ to a high value while ensuring the training accuracy does not fall drastically. We select the model based on the *training accuracy* as higher training accuracy ensures optimal learning of the bias-aligned data points even when there is less learning of the bias-conflicting samples. To avoid all predictions from collapsing to a single class, we

Table 3: **Comparison of different alternatives**. We compare three methods – loss-weighted CE loss (LW), cluster-weighted CE loss (CW), and cluster-based margin loss (CM). While LW leads to improved scores compared to an ERM-trained model, CW further improves upon it for CelebA and ColorMNIST-0.995. Finally, the proposed CM outperforms the above two methods by a large margin. All results are with respect to an ImageNet-pretrained ResNet-18 model.

| Method | Waterbirds | | CelebA | | ColorMNIST-0.995 | | |
|---|---|---|---|---|---|---|---|
| | Worst | Avg | Worst | Avg | Bi-Co | Bi-Al | Avg |
| ERM | 38.90 | 76.22 | 27.20 | 75.43 | 49.09 | 100.00 | 74.55 |
| LW | 69.78 | 83.52 | 45.56 | 79.84 | 50.58 | 99.90 | 75.28 |
| CW | 70.16 | 84.21 | 69.4 | 85.16 | 65.00 | 99.05 | 82.00 |
| CM | **80.29** | **84.56** | **81.61** | **86.04** | **72.56** | 96.28 | **84.42** |

sample equally from each class for each batch of the training set. We call the obtained bias-amplified model $B$, and utilize its knowledge to design a few alternative mitigation strategies. For debiasing, we use the same architecture as that of the biased model $B$, and denote it by $D$.

Before delving into the rest of our method, we discuss a caveat here. An attentive reader might enquire as to whether the above technique of bias amplification will always work. We reiterate that we consider the bias in the downstream dataset to align with that in the feature encoder. An example to the contrary has been discussed in Scenario 1 (subsection 3.2). This presents an interesting future direction where one might attempt to debias a feature encoder which does not capture the downstream dataset bias.

**Technique 1: Loss-weighted (LW) Cross-Entropy Loss.** We use a weighted CE loss to train $D$: $L_{LW}^D = -L_T^B \sum_{j=1}^{|\mathcal{C}|} p_j \log \hat{p}_j$, where $L_T^B$ is the CE loss computed from $B$ for the sample with respect to its ground truth label. The intuition is that $B$ being biased, it would upweight the CE loss for the bias-conflicting sample in the mitigation stage. On implementing this for the Waterbirds, CelebA and ColorMNIST-0.995, we find that even though the mitigation performance increases as compared to the ERM-trained model, the improvement is not satisfactory (see Table 3).

**Technique 2: Cluster-weighted (CW) Cross-Entropy Loss.** The adapter features $\hat{f}$ in the biased model $B$ are expected to encode the bias in the downstream dataset. Hence, we explore a cluster-based weighting scheme for bias mitigation by clustering the bias-amplified feature space $\hat{f}$ in $B$. We define $K$ as the number of clusters obtained and $\bar{m}_c^z$ as the proportion of class $c$ within the cluster $z$, i.e.,

$$\bar{m}_c^z = \frac{n_c^z + \epsilon}{\sum_{c' \in \mathcal{C}} n_{c'}^z + \epsilon} \tag{2}$$

where $n_c^z$ is the number of times samples of class $c$ have occurred in cluster $z$. For any sample in the training data, we find the cluster it belongs to (say $z$), and then weight the CE loss for training $D$ in the following way: $L_{LW}^D = -\sum_{j=1}^{|\mathcal{C}|}(1 - \bar{m}_j^z)p_j \log \hat{p}_j$. Intuitively, the loss is upweighted if the sample belongs to a minority class in its cluster, otherwise it is downweighted. From Table 3, we find that it is a significant improvement over LW. Infact, from Table 2, we find that it performs on par with Co-Ada [50] for Waterbirds and ColorMNIST-0.995.

### 3.5 Cluster-based Adaptive Margin Loss

In Technique 2, we see that clustering-based weighting methods can be effective, where we upweight samples belonging to the minority classes in a cluster, whereas we downweight the others proportionally to increase the importance of the sparse samples. We further ask, inside each cluster, can we make the features of the individual classes more discriminative? We conjecture that in such a case, it will be easier for the debiasing model $D$ to distinguish among the samples from the different groups in the dataset (assuming that the clusters are an approximate representation of the true groups in the dataset). To this end, margin losses are an efficient class of loss functions that can reduce intra-class distance and increase inter-class distance in the feature space. Enforcing such margin constraints on hyperspherical feature spaces has been shown to be beneficial in case of deep face recognition [6, 5, 66], long-tailed learning [67], few-shot learning [68] and the language-bias problem [69, 70] in Visual Question Answering (VQA). Especially, inspired by the works of long-tailed learning and VQA, we

make the margins adaptive to ensure discriminative features among the frequently and infrequently occurring classes within a cluster. To achieve this, we utilize the weights used in the cluster-weighted (CW) CE loss (defined above) as the margin values. We begin by defining the following:

**Normalized CE loss**: First, we reformulate the *CE* loss as a cosine loss [5, 6], by $L2$-normalizing the classifier weight vectors $\mathbf{W_k} \in C$ (recall that $C$ is the classifier) for each class $c_k \in \mathcal{C}$ ($k = 1, 2, \ldots |\mathcal{C}|$), and the trainable feature $\hat{f}$. We define $\hat{\mathbf{W}}_{\mathbf{k}} = \frac{\mathbf{W_k}}{\|\mathbf{W_k}\|}$ and $\hat{\mathbf{f}}_{\mathbf{norm}} = s\frac{\hat{\mathbf{f}}}{\|\hat{\mathbf{f}}\|}$, where $s$ is a scaling parameter. The bias term is set to $0$ for simplicity. Let $\theta_k$ be the angle between $\hat{\mathbf{f}}$ and $\mathbf{W_k}$. Therefore, the logit for each class $c_k$ becomes: $\hat{y}_k = \hat{\mathbf{W}}_{\mathbf{k}}^{\top}\hat{\mathbf{f}}_{\mathbf{norm}} = \|\hat{\mathbf{W}}_{\mathbf{k}}\|\|\hat{\mathbf{f}}_{\mathbf{norm}}\| \cos\theta_k = s \cos\theta_k$. The features $\hat{\mathbf{f}}_{\mathbf{norm}}$ are thus distributed on a hypersphere with a radius $s$. This makes the normalized *CE* loss for a single sample as:

$$L_{NS} = \sum_{k=1}^{|\mathcal{C}|} -p_k \log \frac{\exp(s \ \cos\theta_k)}{\sum_{j=1}^{|\mathcal{C}|} \exp(s \ \cos\theta_j)} \tag{3}$$

**Adaptive Margin Loss.** Inspired by the ArcFace loss [6] used in face recognition, we define the adaptive margin loss here. The loss adds a margin penalty to the angle between the features $\hat{\mathbf{f}}$, and the classifier weights $\mathbf{W_k}$ for the $k^{th}$ class. Since the margin is placed on the angle, it maps exactly to the "geodesic" distance on the hypersphere [6], leading to highly discriminative features. For face recognition, it suffices to have a constant value for the margin penalty ($0.5$ for ArcFace). Since we want discriminative features for the frequently and infrequently occurring classes in a cluster, we ensure that the training samples assigned to a cluster $z$ and belonging to the majority class $y$ in that cluster are allowed a smaller margin than those belonging to the minority class. We assign the loss weights from CW as the margin values (i.e. eq. 2). Specifically, for a data sample belonging to class $c$ with cluster id $z$, we denote its margin as $m_c^z = (1 - \bar{m}_c^z)$ (see Fig. 3 for an overview of the system). Finally, we define the *angular* adaptive margin loss for each sample belonging to the tuple $t = (c, z)$ by combining eq.s 3 and 2:

$$L_{\text{Margin}}^t = \sum_{k=1}^{|C|} -p_k \log \frac{\exp(s \ \cos(\theta_k + m_{c_k}^z))}{\sum_{j=1}^{|C|} \exp(s \ \cos(\theta_j + m_{c_j}^z))} \tag{4}$$

**Gaussian Randomization of the margins**: While we estimate the margins for the margin loss by clustering the features from the biased model $B$, the obtained clusters can capture noisy signals, leading to erronous results. Alluding to our previous example of a dataset of people with blond and non-blond hair with gender bias (Section 1), since most of the females have blond hair, all such females should be given lesser margin values, whereas blond-haired males should be given higher values. But clustering leads to noisy grouping of samples in the dataset, whereby a certain group may include less blond females and more blond males. Therefore, for those female blond samples, comparatively large margin penalty will be assigned, while the blond male samples will get a low margin penalty. Also, the margins being high for sparse classes and low for the frequent ones in a cluster, they may over-learn the former, ignoring the latter. In face recognition, Boutros et al. [71] suggest that in typical margin losses, setting constant margins can limit the generalizability and discriminative power of a model and advise the introduction of stochasticity in the margin values to boost the same. This stochasticity can help us smooth out the effect of the noises in clustering on one hand and make the model more generalized to all groups. To enforce this, we use a randomized version of $\bar{m}_c^z$, called $\bar{r}_c^z$, where $\bar{r}_c^z \sim \mathcal{N}(\bar{m}_c^z, \sigma)$. Recall that $\bar{m}_c^z$ denotes the proportion of samples belonging to class $c$ and cluster $z$. $\mathcal{N}$ is the Gaussian distribution, and $\sigma$ is the standard deviation (a hyperparameter). This impedes the model from overcorrecting with respect to the rare classes in a group, thus increasing its generalizability while also compensating for the errors in margin values due to the noisy cluster labels. Finally, we obtain the randomized margin $r_c^z = 1 - \bar{r}_c^z$ for each $c \in \mathcal{C}$. We then replace $m_c^z$ with $r_c^z$ in eq. 4. Our overall method is summarized in Fig. 3.

## 4 Experiments and Results

**Dataset Details**. We evaluate our method on multiple benchmarks. **Waterbirds** [23] is a dataset of birds, labeled as *waterbird* if the bird is a seabird, and *landbird* otherwise. A spurious correlation exists between these labels and the background–land or water. The dataset has $4795$ training samples. The **ColorMNIST** dataset (CMNIST) [72] is generated from MNIST [73], where each digit is

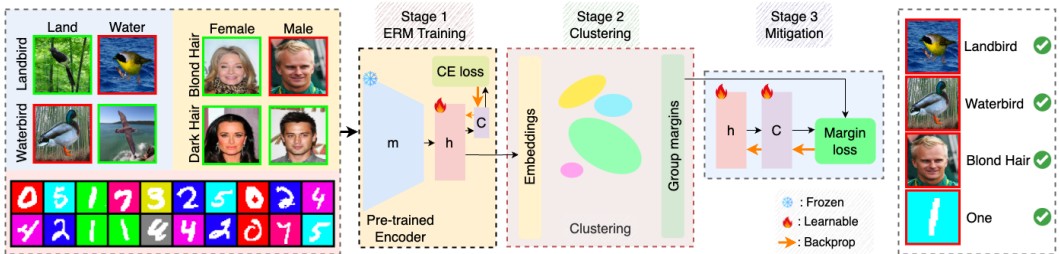

Figure 3: **Overview of our method.** There are 3 steps: a) **Bias-amplified training** of the model through Cross-Entropy Loss with high weight decay, b) **Clustering** the biased features $\hat{f}$, c) **Mitigating** the biases by using the resultant clusters to calculate the margins and the corresponding loss, leading to decent performance on the bias-conflicting data points.

Table 4: **Performance comparison of our method (CM)**. Worst and Average Group Accuracies for Waterbirds & CelebA (Blond Hair), Bias-Conflicting (Bi-Co), Bias-Aligned (Bi-Al) and Average Accuracies for CMNIST-0.9. Highest accuracies are marked in bold, the $2^{nd}$ highest ones are underlined. All scores for our method are averaged over 3 seeds.

| Backbone | Model | Waterbirds | | CelebA | | CMNIST-0.9 | | |
|---|---|---|---|---|---|---|---|---|
| | | Worst | Avg | Worst | Avg | Bi-Co | Bi-Al | Avg |
| ResNet-18 | ERM | $38.90^{\pm1.40}$ | $76.22^{\pm1.04}$ | $27.20^{\pm0.89}$ | $75.43^{\pm0.67}$ | $61.72^{\pm0.59}$ | $99.90^{\pm0.31}$ | $65.54^{\pm0.27}$ |
| | Co-Ada [50] | $\underline{67.57}^{\pm1.29}$ | $\underline{80.10}^{\pm1.36}$ | $\underline{78.37}^{\pm0.14}$ | $\underline{85.79}^{\pm0.78}$ | $\underline{80.82}^{\pm1.02}$ | $89.63^{\pm0.11}$ | $\underline{85.22}^{\pm0.37}$ |
| | CM (Ours) | $\mathbf{80.29}^{\pm2.50}$ | $\mathbf{84.56}^{\pm1.20}$ | $\mathbf{81.61}^{\pm1.02}$ | $\mathbf{86.04}^{\pm0.26}$ | $\mathbf{81.91}^{\pm0.40}$ | $\underline{92.75}^{\pm0.56}$ | $\mathbf{87.33}^{\pm0.35}$ |
| CLIP RN50 | ERM | $67.93^{\pm1.12}$ | $84.65^{\pm0.71}$ | $36.09^{\pm0.42}$ | $79.59^{\pm0.70}$ | $90.22^{\pm0.12}$ | $99.18^{\pm1.04}$ | $91.18^{\pm0.05}$ |
| | Co-Ada [50] | $\mathbf{81.95}^{\pm1.13}$ | $\mathbf{87.35}^{\pm0.85}$ | $\mathbf{90.52}^{\pm0.17}$ | $\underline{91.88}^{\pm1.15}$ | $83.81^{\pm0.02}$ | $95.68^{\pm0.01}$ | $89.74^{\pm0.15}$ |
| | CM (Ours) | $\underline{79.28}^{\pm0.94}$ | $\underline{85.79}^{0.46}$ | $\underline{90.47}^{\pm0.43}$ | $\mathbf{92.52}^{\pm0.42}$ | $\mathbf{94.07}^{\pm0.06}$ | $\mathbf{96.40}^{\pm0.06}$ | $\mathbf{95.23}^{\pm0.03}$ |
| ViT-B | ERM | $59.67^{\pm1.07}$ | $\mathbf{83.31}^{\pm0.89}$ | $31.72^{\pm0.64}$ | $77.82^{\pm0.62}$ | $\underline{88.93}^{\pm0.4}$ | $\mathbf{99.69}^{\pm0.15}$ | $\underline{94.31}^{\pm0.14}$ |
| | Co-Ada [50] | $\underline{63.71}^{\pm1.18}$ | $80.24^{\pm1.15}$ | $\underline{85.87}^{\pm0.42}$ | $\mathbf{89.45}^{\pm1.17}$ | $79.71^{\pm0.61}$ | $86.84^{\pm0.07}$ | $83.28^{\pm1.01}$ |
| | CM (Ours) | $\mathbf{74.92}^{\pm1.43}$ | $\underline{82.85}^{\pm0.34}$ | $\mathbf{87.28}^{\pm0.91}$ | $\underline{89.31}^{\pm0.66}$ | $\mathbf{91.32}^{\pm0.19}$ | $\underline{98.59}^{\pm0.15}$ | $\mathbf{94.96}^{\pm0.16}$ |

associated with one color in its background most of the time (see Fig. 3). It has 50000 training and 10000 test images. We evaluate our margin loss method on two variants of CMNIST: in CMNIST-0.9, each digit is associated with one color 90% of the time and other colors only 10% of the time (moderate bias). In CMNIST-0.995, each digit is associated with a single color 99.5% of the time (severe bias). The real-world **CelebA** dataset [64] consists of $202,599$ face images of celebrities along with 40 attributes. We choose *Blond Hair* as the target attribute, as it is known to suffer from severe gender bias [28, 38, 45]. For evaluating models on CelebA and Waterbirds, we find the accuracy of each group $g = (c, a)$ in the test set [20, 45], and report the worst of all the groups in $\mathcal{G}$ and their average. For CMNIST, we report the overall bias-aligned (i.e. images of digits with their maximally associated colors) and bias-conflicting (i.e. images of digits with other colors) accuracies, along with the average of all the groups [38]. We also show the results for two more real-life datasets (BAR [20] and UTKFace [74]) in Appendix Table 12.

**Implementation Details**. We evaluate our method on multiple pretrained encoders: ImageNet-pretrained ResNet-18 [75] and ViT-Base [76] encoders, and CLIP ResNet-50 image encoder [12] (pretrained on other datasets). All our implementations use a 1-hidden layer of $M$ neurons followed by a non-linearity in the adapter, and the classifier is a linear layer. For clustering, we use the KMeans algorithm. We discuss the hyperparameters and their analyses in Appendix subsection A.3. We assume the availability of a small group-balanced (but unannotated) validation set and calculate the overall accuracy over this dataset for model selection during the bias-mitigation phase.

## 4.1 Results

We compare our baseline against Contrastive Adapter [50], given that it performs the best among all other methods across different benchmarks (see Table 2), and the ERM model. The results are presented in Table 4, where we compare our method against the ERM method and Co-Ada. For comparison with other competing methods, see Appendix Table 8. We also present the results for the ViT-H 14 encoder in Appendix Table 13.

Table 6: **Ablations of the proposed method**: Here we show the roles of the different components of our model using ResNet-18 as the pretrained backbone.

| Model Component | Waterbirds | | CelebA | | CMNIST-0.9 | | |
| --- | --- | --- | --- | --- | --- | --- | --- |
| | Worst | Average | Worst | Average | Bi-Co | Bi-Al | Average |
| CM (Ours) | 80.29 | 84.56 | 81.61 | 86.04 | 81.91 | 92.75 | 87.33 |
| Constant Margin (0.5) | 55.76 | 77.45 | 34.44 | 77.63 | 44.91 | 100.00 | 72.45 |
| No Randomization | 79.28 | 82.9 | 78.51 | 86.07 | 82.28 | 91.23 | 86.70 |
| Clustering from $f^{\text{locked}}$ | 74.92 | 84.95 | 80.52 | 85.04 | 75.96 | 94.88 | 85.42 |

For the *Waterbirds* dataset, our method far outperforms Contrastive Adapter for ResNet-18 and ViT-B (by 12.72% and 11.21% respectively), though for CLIP the worst-group accuracy is slightly lesser (by 2.67%). In case of *CelebA*, for ResNet-18 and ViT-Base, the margin loss outperforms Contrastive Adapter (by 3.24% and 1.41%). For the CLIP encoder, we observe the two methods to perform similarly. The *CMNIST-0.9* dataset is affected by a moderate degree of the background color bias, whereas for *CMNIST-0.995* the bias strength is severe. In case of both datasets, our method outperforms Contrastive Adapter for all backbones by a large margin, especially for CMNIST-0.9 (see results for CMNIST-0.995 in Table 5). Compared to Contrastive Adapter, our method has another advantage: it is *time-efficient*, as discussed in the Appendix Table 17. Our method is effective even when finetuning to the encoder weights is possible (see Appendix subsection A.7).

## 5 Ablations

In this subsection, we discuss the important components of our approach and the margin loss strategy. We first show what happens when the margin penalty is constant as the ArcFace loss [6] itself. Then, we show the role played by the Gaussian randomization of the group-based margins (read subsection 3.5 and Appendix subsection **??**). Finally, we show what happens when we estimate the different bias-groups in the training data by clustering the blackbox features instead of the bias-amplified adapter layer. All evaluations are performed on Waterbirds, CelebA (Blond Hair classification) and CMNIST-0.9 with ResNet-18 as the backbone. The results are shown in Table 6.

**Constant Margin**. We present the results obtained by keeping the margin value constant at 0.5 as suggested by ArcFace [6] (no Gaussian randomization is applied here). This study en-

Table 5: **Performance comparison of our method for CMNIST-0.995**. We report Bias-Conflicting (Bi-Co), Bias-Aligned (Bi-Al) and Average Accuracies. Highest accuracies are marked in bold, the $2^{nd}$ highest ones are underlined. All scores for our method are averaged over 3 seeds.

| | Model | CMNIST-0.995 | | |
| --- | --- | --- | --- | --- |
| | | Bi-Co | Bi-Al | Avg |
| R18 | ERM | $49.09^{\pm0.19}$ | $100.00^{\pm0.70}$ | $74.55^{\pm0.82}$ |
| | Co-Ada | $\underline{65.48}^{\pm0.21}$ | $84.48^{\pm0.42}$ | $\underline{74.98}^{\pm0.49}$ |
| | CM (Ours) | $\mathbf{72.56}^{\pm0.88}$ | $\underline{96.28}^{\pm0.50}$ | $\mathbf{84.42}^{\pm0.41}$ |
| CLIP | ERM | $56.87^{\pm0.12}$ | $100.00^{\pm0.46}$ | $78.44^{\pm0.52}$ |
| | Co-Ada | $\underline{77.34}^{\pm0.34}$ | $92.18^{\pm0.06}$ | $\underline{84.76}^{\pm0.18}$ |
| | CM (Ours) | $\mathbf{80.65}^{\pm0.05}$ | $\underline{93.29}^{\pm1.23}$ | $\mathbf{86.97}^{\pm0.60}$ |
| ViT-B | ERM | $59.21^{\pm0.65}$ | $100.00^{\pm0.17}$ | $79.61^{\pm0.42}$ |
| | Co-Ada | $\underline{76.88}^{\pm0.43}$ | $92.27^{\pm0.61}$ | $\underline{84.58}^{\pm0.70}$ |
| | CM (Ours) | $\mathbf{77.97}^{\pm0.71}$ | $\underline{94.86}^{\pm0.49}$ | $\mathbf{86.42}^{\pm0.44}$ |

ables us to judge the utility of the adaptive nature of the margins. While for Waterbirds and Blond Hair, the scores are better than those of ERM, there's a sharp drop in both worst and average group accuracies compared to our method. The trends are similar in CMNIST-0.9 as well. Hence, we conclude that while margin losses improve model performance, a constant margin is not enough.

**Without Randomization**. We remove the Gaussian randomization of the margins, but here the margins are adaptive (as per eq. 2). For Waterbirds, we see that both the worst and average group accuracies reduce compared to the final scores. Worst group accuracy drops for CelebA as well, though the average group score remains similar. In CMNIST, the bias-conflicting accuracies slightly improve at the cost of a small drop in bias-aligned performance. Thus, overall we find that randomizing the margins for each sample is helpful.

**Clustering from $f^{\text{locked}}$**. Our model identifies biases by clustering the adapter features $\hat{f}$. However, the pretrained features $f_{\text{locked}}$ can be useful as well, if the pretrained model and the downstream

dataset share similar biases. Hence, we cluster the pretrained features $f^{\text{locked}}$ instead of $\hat{f}$ to obtain the margin penalties. The rest of the training pipeline remains the same, as described in subsection 3.3. For all 3 datasets, we see a reduction in the worst and average group accuracies, however, the scores are still close to that of the proposed approach. Since clustering the pretrained features does not require an extra stage of ERM training, therefore in presence of time-constraints, $f^{\text{locked}}$ can be used as a proxy for $\hat{f}$, and subsequently, relatively decent model predictions can be obtained.

## 6   Discussion and Limitations

Our method is specifically targeted towards cases where the bias in the downstream dataset is already encoded in the pretrained model. We believe that detecting if this assumption holds apriori is highly challenging in the absence of the bias labels. We put forward a few suggestions to identify the scenarios that fit this assumption: a) Obtain bias annotations for the small validation set. If the worst group accuracy of the validation set does not reduce substantially with increasing weight decay, it indicates that the features have stronger signals of the target class than that of the bias, making it harder to capture the bias. b) If the bias annotations of the validation set cannot be obtained due to privacy concerns, the overall validation accuracy can indicate strength of the bias. For example, the difference between the validation and training accuracy is $25\%$ for an ERM trained method for the Waterbirds dataset on the ResNet-18 backbone. The higher this difference, the more the indication that the model is overfitting to more and more samples. Such overfitting can indicate that the model is learning the bias in the dataset, thus not generalizing on the bias-conflicting samples. While we have mentioned the situation when the pretrained model is aware of the target attribute in Section 3, another potential use case might be when the pre-training data distribution may be different from that of the downstream dataset. We prescribe finetuning the pretrained models to amplify and mitigate the biases in such a case. A more practical approach would be to choose another model, as also suggested by Zhang et al. [50].

One limitation of our method is that it currently relies on the existence of biased groups in the training set (i.e., the bias labels are expected to be categorical) as it uses clustering to identify the biases. Furthermore, since our approach (along with most competing methods) relies on amplifying the bias first and then mitigating the same, it involves a risk wherein if the mitigation module fails, the bias in the system may be exacerbated. With these limitations in mind, we hope this work initiates a much required discussion in this direction leading to more sophisticated and targeted solutions in the near future.

**Scalability of the proposed approach to large datasets and models**. Since our method involves clustering the features, if the dataset is large-scale, one can randomly sample a small percentage of it to do the clustering. For example, on clustering only $10\%$ randomly sampled images from the CelebA dataset, we find that the worst group accuracy is $81.11\%$, whereas the average group accuracy is $85.6\%$ on the ResNet-18 backbone. The scores become $80.55\%$ and $85.6\%$ respectively when the clustering is performed only on $1\%$ of the images (with full clustering, the scores are $81.61\%$ and $86.04\%$ respectively). On the other hand, for very large models, the requirement is to be able to load the model into a GPU memory. Our method adds negligible overhead owing to the addition of only an adapter and a classifier layer.

## 7   Conclusion

In this work, we explored the effect of using pretrained but frozen feature extractors on downstream applications with biased datasets and found the need for specific bias mitigation strategies in cases where the biases in the downstream dataset align with the pretrained encoders. Such mitigation is challenging as the encoder is blackbox. While we found many of the existing works to be inadequate, we proposed a simple method where we first amplified the biases present in the downstream dataset and then employed a clustering-based margin loss to mitigate the same. Our experiments showed this method to be an efficient and effective technique across multiple benchmarks. Lastly, it is our hope that this work will initiate a much-required discussion among the scientific community on encouraging similar works in this highly practical problem setting. We share further studies, detailed hyperparameter analyses and experiments of the proposed method in the appendix.

# 8 Acknowledgements

This research is funded by the Qualcomm Innovation Fellowship. Abhipsa Basu is supported by the PMRF Fellowhsip.

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

# A   Appendix

## A.1   Biases in Pretrained Features - Further Analysis

In the main paper, we looked at two scenarios on how the absence or presence of biases in the pretrained feature encoder affects the downstream model's performance on finetuning data. However, for the analysed cases, we did not look into the specific relation between the pretraining data and the downstream data. In the appendix, we perform a controlled and detailed analysis to answer 4 questions in this regard:

**Q1: What happens if the pretraining data is heavily biased and so is the downstream dataset?**

To answer this question, we pretrain a ResNet-18 model with the CelebA dataset to make it heavily gender-biased by performing gender prediction itself as the pretraining task [14]. During the downstream step, we choose to predict the Blond Hair attribute of the CelebA dataset, which is known to be heavily correlated with gender. We *class-balance* the finetuning dataset to have equal number of blonds and non-blonds, so as to avoid the downstream predictions from collapsing into a single class. Finally, we also sample from the class-balanced downstream data to ensure that the total number of data points in the downstream data is approximately $1/4^{\text{th}}$ of the pretraining data.

**Q2: What happens if the pretraining dataset is heavily biased but the downstream dataset is less biased?**

Table 7: Dependence of downstream model performance on pretraining data. When the pretraining model has a high amount of bias, the downstream data performance is heavily affected if the latter itself is biased. On the other hand, the performance is more stable if i) the pretraining data has high bias and the finetuning data has low bias, ii) the pretraining data has low bias and the finetuning data has high bias. The downstream predictions get biased even if the finetuning data has biases which are not common with those in the pretraining data, but the performance drops.

| Pretraining data | Finetuning Data | Worst-Group | Average-Group |
|---|---|---|---|
| High Bias | High Bias | 6.48 | 53.21 |
| High Bias | Low Bias | 43.33 | 57.97 |
| Low Bias | High Bias | 46.28 | 54.95 |
| High Bias | High Bias (WB) | 4.70 | 49.76 |

For pretraining, we use gender prediction again as mentioned in the above case, but for the downstream dataset, we *group-balance* the CelebA dataset in terms of Blond Hair and gender, keeping the total number of downstream samples same as described above.

**Q3: What happens if the pretraining data is less biased but the downstream dataset is heavily biased?**

For the downstream dataset, we choose the same one as mentioned in case of Q1. For the pretraining task, instead of predicting gender, we predict the 'Smiling' attribute, as we find it to be much less biased in terms of gender.

**Q4: What happens if the pretraining data is highly biased and so is the downstream dataset, but the two biases are unrelated?**

To simulate this scenario, we pretrain the ResNet-18 encoder by predicting the gender attribute again in the CelebA dataset as mentioned in the above cases, and for the downstream task, we choose a *class-balanced* version of Waterbirds, which has a completely different bias.

**Findings.** Note that Q1, Q2 and Q3 are related. From our experiments, we observe the following. When both the pretrained encoder and the downstream dataset are heavily biased with a common attribute (Q1), the worst group performance on the downstream dataset is the least of all the 3 cases. On the other hand, for Q2 (i.e. when the pretrained model is heavily biased but the downstream dataset is not), the worst group performance improves considerably. This corroborates with our findings in the main paper (Sec 3) that group-balancing the downstream task can improve model performance. The best performance is seen for Q3, where the pretraining model is less biased, but the downstream dataset is still highly biased – here we find that the worst group performance is infact higher than the previous case (Q2). This shows that the lesser the bias in the pretraining data, the more unbiased downstream application is going to be.

When the bias in the pretrained model and the downstream task are unrelated, we find that the model is indeed affected by the bias in the downstream data, with a drastic drop in overall performance. This indicates that if the pretraining and downstream data distributions do not match, one should choose a different feature encoder to achieve better performance on the downstream task. All the above findings are summarized in Table 7. Note that the overall performance of the downstream task is considerably lower in comparison to using an Imagenet pretrained encoder for all above cases. This is likely because the pretraining task involves a binary prediction of an attribute that is different from the downstream target attribute – leading to overfitting of the pretrained features to the former.

### A.2  Comparison of the Margin Loss-based method with other Methods

In the main paper, we discuss the performance of our proposed baseline and compare it with Co-Ada [50] across various benchmarks and pretrained encoders. In this section, we describe the performance of the other methods in Table 8. We find that for the ResNet-18 encoder, among all methods, LfF [20] performs well on Waterbirds and both the versions of CMNIST, but the worst group accuracies for the same method is poor for CelebA. On the other hand, while BPA has high worst group accuracy for CelebA, its performance deteriorates for Waterbirds and both the CMNIST versions. This inconsistency is present for other methods as well, for every encoder.

Table 8: **Performance comparison of our method with all other methods**. Worst and Average Group Accuracies for Waterbirds & CelebA (Blond Hair), Bias-Conflicting (Bi-Co), Bias-Aligned (Bi-Al) and Average Accuracies for CMNIST-0.9 and CMNIST-0.995. Highest accuracies are marked in bold, while the $2^{nd}$ highest ones are underlined. All scores with respect to the proposed approach are averaged over 3 seeds. Co-Ada: Contrastive Adapter.

| Backbone | Model | Waterbirds | | CelebA | | CMNIST-0.9 | | | CMNIST-0.995 | | |
| | | Worst | Avg | Worst | Avg | Bi-Co | Bi-Al | Avg | Bi-Co | Bi-Al | Avg |
|---|---|---|---|---|---|---|---|---|---|---|---|
| ResNet-18 | ERM | $38.90^{\pm1.40}$ | $76.22^{\pm1.04}$ | $27.20^{\pm0.89}$ | $75.43^{\pm0.67}$ | $61.72^{\pm0.59}$ | $99.90^{\pm0.31}$ | $65.54^{\pm0.27}$ | $49.09^{\pm0.19}$ | $100.00^{\pm0.70}$ | $74.55^{\pm0.82}$ |
| | DebiAN [38] | $58.94^{\pm1.09}$ | $80.47^{\pm0.91}$ | $26.10^{\pm0.02}$ | $75.41^{\pm0.17}$ | $61.97^{\pm0.64}$ | $99.72^{\pm0.47}$ | $80.84^{\pm0.19}$ | $49.82^{\pm0.82}$ | $99.80^{\pm0.11}$ | $74.81^{\pm0.27}$ |
| | BPA [45] | $58.70^{\pm2.49}$ | $80.83^{\pm0.94}$ | $66.71^{\pm0.70}$ | $84.14^{\pm0.66}$ | $61.43^{\pm0.58}$ | $98.01^{\pm0.36}$ | $65.09^{\pm0.35}$ | $47.84^{\pm0.79}$ | $100.00^{\pm0.21}$ | $73.92^{\pm0.22}$ |
| | LfF [20] | $66.09^{\pm1.21}$ | $\underline{81.39^{\pm0.81}}$ | $13.26^{\pm0.31}$ | $69.42^{\pm0.43}$ | $\underline{78.46^{\pm0.32}}$ | $84.15^{\pm0.43}$ | $\underline{81.30^{\pm0.71}}$ | $\underline{71.61^{\pm0.32}}$ | $91.47^{\pm0.04}$ | $\underline{81.54^{\pm0.05}}$ |
| | JTT [19] | $49.84^{\pm1.21}$ | $77.03^{\pm0.86}$ | $56.25^{\pm0.45}$ | $73.58^{\pm0.36}$ | $47.82^{\pm0.37}$ | $95.35^{\pm0.64}$ | $71.59^{\pm0.43}$ | $42.86^{\pm0.07}$ | $99.79^{\pm0.31}$ | $71.32^{\pm0.43}$ |
| | GEORGE [44] | $59.35^{\pm2.21}$ | $80.34^{\pm0.86}$ | $42.22^{\pm0.41}$ | $79.76^{\pm0.34}$ | $61.41^{\pm0.65}$ | $99.90^{\pm0.84}$ | $80.65^{\pm0.62}$ | $48.77^{\pm0.39}$ | $100.00^{\pm0.67}$ | $74.38^{\pm0.09}$ |
| | Co-Ada [50] | $\underline{67.57^{\pm1.29}}$ | $80.10^{\pm1.36}$ | $\underline{78.37^{\pm0.14}}$ | $\underline{85.79^{\pm0.78}}$ | $80.82^{\pm1.02}$ | $89.63^{\pm0.11}$ | $85.22^{\pm0.37}$ | $65.48^{\pm0.21}$ | $84.48^{\pm0.42}$ | $74.98^{\pm0.49}$ |
| | Our method | $\mathbf{80.29^{\pm2.50}}$ | $\mathbf{84.56^{\pm1.20}}$ | $\mathbf{81.61^{\pm1.20}}$ | $\mathbf{86.04^{\pm0.26}}$ | $\mathbf{81.91^{\pm0.40}}$ | $92.75^{\pm0.56}$ | $\mathbf{87.33^{\pm0.35}}$ | $\mathbf{72.56^{\pm0.88}}$ | $96.28^{\pm0.49}$ | $\mathbf{84.42^{\pm0.44}}$ |
| CLIP RN50 | ERM | $67.93^{\pm1.12}$ | $84.65^{\pm0.71}$ | $36.09^{\pm0.42}$ | $79.59^{\pm0.70}$ | $90.22^{\pm0.12}$ | $\underline{99.18^{\pm1.04}}$ | $91.18^{\pm0.05}$ | $56.87^{\pm0.12}$ | $100.00^{\pm0.46}$ | $78.44^{\pm0.52}$ |
| | DebiAN [38] | $63.61^{\pm1.78}$ | $84.73^{\pm0.78}$ | $38.33^{\pm0.65}$ | $79.98^{\pm0.39}$ | $90.62^{\pm0.51}$ | $\mathbf{99.20^{\pm0.67}}$ | $\underline{94.91^{\pm0.34}}$ | $58.29^{\pm0.61}$ | $99.90^{\pm0.19}$ | $79.10^{\pm0.24}$ |
| | BPA [45] | $69.51^{\pm2.19}$ | $85.71^{\pm1.21}$ | $81.70^{\pm0.75}$ | $90.09^{\pm0.47}$ | $91.61^{\pm0.64}$ | $98.89^{\pm0.39}$ | $92.34^{\pm0.47}$ | $55.24^{\pm0.34}$ | $100.00^{\pm0.63}$ | $77.62^{\pm0.27}$ |
| | LfF [20] | $61.28^{\pm1.42}$ | $80.07^{\pm0.79}$ | $37.57^{\pm0.46}$ | $79.63^{\pm0.82}$ | $91.09^{\pm0.67}$ | $96.08^{\pm0.39}$ | $93.59^{\pm0.07}$ | $\underline{80.17^{\pm0.41}}$ | $81.95^{\pm0.60}$ | $81.06^{\pm0.02}$ |
| | JTT [19] | $71.18^{\pm2.01}$ | $83.72^{\pm1.41}$ | $75.00^{\pm0.45}$ | $84.78^{\pm0.52}$ | $90.61^{\pm0.06}$ | $97.82^{\pm0.18}$ | $94.22^{\pm1.07}$ | $52.39^{\pm0.34}$ | $98.37^{\pm0.28}$ | $75.38^{\pm0.09}$ |
| | GEORGE [44] | $62.77^{\pm1.21}$ | $83.14^{\pm0.89}$ | $56.11^{\pm0.65}$ | $85.07^{\pm0.27}$ | $90.61^{\pm0.41}$ | $98.47^{\pm0.09}$ | $94.54^{\pm0.44}$ | $54.65^{\pm0.63}$ | $100.00^{\pm0.00}$ | $77.32^{\pm0.03}$ |
| | Co-Ada [50] | $\mathbf{81.95^{\pm1.13}}$ | $\mathbf{87.35^{\pm0.85}}$ | $90.52^{\pm17}$ | $\underline{91.88^{\pm1.15}}$ | $83.81^{\pm0.02}$ | $95.68^{\pm0.01}$ | $89.74^{\pm0.15}$ | $77.34^{\pm0.34}$ | $92.18^{\pm0.06}$ | $\underline{84.76^{\pm0.18}}$ |
| | Our method | $\underline{79.28^{\pm0.94}}$ | $\underline{85.79^{\pm0.46}}$ | $\underline{90.47^{\pm0.43}}$ | $\mathbf{92.52^{\pm0.42}}$ | $\mathbf{94.07^{\pm0.06}}$ | $96.40^{\pm0.06}$ | $\mathbf{95.23^{\pm0.03}}$ | $\mathbf{80.65^{\pm0.05}}$ | $93.29^{\pm1.23}$ | $\mathbf{86.97^{\pm0.60}}$ |
| ViT-B | ERM | $59.67^{\pm1.07}$ | $\mathbf{83.31^{\pm0.89}}$ | $31.72^{\pm0.64}$ | $77.82^{\pm0.62}$ | $88.93^{\pm0.4}$ | $\mathbf{99.69^{\pm0.15}}$ | $94.31^{\pm0.14}$ | $59.21^{\pm0.65}$ | $100.00^{\pm0.70}$ | $79.61^{\pm0.82}$ |
| | DebiAN [38] | $58.32^{\pm1.82}$ | $\underline{82.88^{\pm0.87}}$ | $29.40^{\pm0.71}$ | $77.05^{\pm0.62}$ | $89.54^{\pm0.45}$ | $99.6^{\pm0.32}$ | $\underline{94.57^{\pm0.81}}$ | $60.63^{\pm0.09}$ | $99.91^{\pm0.26}$ | $80.26^{\pm0.64}$ |
| | BPA [45] | $59.01^{\pm1.45}$ | $82.03^{\pm0.73}$ | $60.00^{\pm0.61}$ | $84.93^{\pm0.49}$ | $\underline{89.28^{\pm0.82}}$ | $\mathbf{99.69^{\pm0.64}}$ | $94.49^{\pm0.39}$ | $57.45^{\pm0.29}$ | $100.00^{\pm0.61}$ | $78.73^{\pm0.43}$ |
| | LfF [20] | $30.02^{\pm1.83}$ | $73.55^{\pm0.69}$ | $28.73^{\pm0.72}$ | $74.48^{\pm0.62}$ | $89.23^{\pm0.43}$ | $\mathbf{99.69^{\pm0.61}}$ | $92.81^{\pm0.43}$ | $76.74^{\pm0.08}$ | $93.58^{\pm0.43}$ | $\underline{85.16^{\pm0.32}}$ |
| | JTT [19] | $52.64^{\pm1.47}$ | $78.84^{\pm0.89}$ | $65.12^{\pm0.52}$ | $78.97^{\pm0.62}$ | $88.28^{\pm0.37}$ | $99.43^{\pm0.45}$ | $93.36^{\pm0.59}$ | $58.41^{\pm0.42}$ | $99.67^{\pm0.36}$ | $79.04^{\pm0.08}$ |
| | GEORGE [44] | $58.72^{\pm1.45}$ | $81.67^{\pm0.87}$ | $55.00^{\pm0.19}$ | $84.14^{\pm0.61}$ | $87.43^{\pm0.43}$ | $98.88^{\pm0.28}$ | $93.15^{\pm0.35}$ | $57.25^{\pm0.73}$ | $100.00^{\pm0.00}$ | $78.62^{\pm0.04}$ |
| | Co-Ada [50] | $\underline{63.71^{\pm1.18}}$ | $80.24^{\pm1.15}$ | $\underline{85.87^{\pm0.42}}$ | $\mathbf{89.45^{\pm1.17}}$ | $79.71^{\pm0.61}$ | $86.84^{\pm0.07}$ | $83.28^{\pm1.01}$ | $\underline{76.88^{\pm0.43}}$ | $92.27^{\pm0.61}$ | $84.58^{\pm0.70}$ |
| | Our method | $\mathbf{74.92^{\pm1.43}}$ | $82.85^{\pm0.34}$ | $\mathbf{87.28^{\pm0.91}}$ | $\underline{89.31^{\pm0.66}}$ | $\mathbf{91.32^{\pm0.19}}$ | $98.59^{\pm0.15}$ | $\mathbf{94.96^{\pm0.16}}$ | $\mathbf{77.97^{\pm0.71}}$ | $94.86^{\pm0.49}$ | $\mathbf{86.42^{\pm0.44}}$ |

## A.3 Hyperparameter Details and Analysis

As mentioned in the main paper, for model selection, we use the training accuracy in the bias-amplification stage, whereas for the mitigation stage, we use the overall validation accuracy. Hyperparameter tuning happens via the validation accuracy of the model checkpoint saved during the mitigation stage. The bias-amplification stage has the following hyperparameters: LR (learning rate), BS (batch size), $\lambda$ (weight decay) and number of epochs. In the clustering stage, the number of clusters $K$ is a hyperparameter. Finally, in the mitigation stage, we have two hyperparameters specific to the margin loss: the scaling parameter $s$ (i.e. the radius of the hypersphere on which the features are projected) and the standard deviation for the Gaussian randomization, $\sigma$. We next define the range of each hyperparamter on which we evaluate our model and the other methods. For learning rate LR, we explore in the range $0.0001, 0.0005, \cdots, 0.05$ in step sizes of 5 and 2 respectively. Similarly, for batch size BS we evaluate on the range $\{64, 128, 256, 512\}$. We explore higher values of weight decay $\lambda \in \{1, 0.1, 0.05, 0.01\}$ for the bias-amplication stage and for the mitigation stage, we search $\lambda$ in the range $10^{-6}, \cdots, 10^{-2}$ in step sizes of 10 and also consider $\lambda = 0$. Number of epochs for training is kept in the range $\{50, 100\}$. For the clustering stage, we choose $K \in \{2, 4, 6, 8\}$ for Waterbirds and CelebA, and explore $K = 10, 20, \cdots, 60$ with a step size of 10 for the CMNIST variants. In the mitigation stage, we select $s \in \{4, 8, 12, 16\}$ and $\sigma \in \{0, 0.05, 0.1, 0.15, 0.2\}$. We show the selected hyperparameters in the bias-amplification stage in Tables 9 and 10, and for the mitigation procedure, the chosen hyperparameters are present in Table 11. For number of neurons $M$ in the MLP layer, we fix the value to be 128. For CelebA experiments, we use SGD as the optimizer, whereas for Waterbirds, and both the variants of CMNIST, we use Adam.

Finally, we show the changes in scores by varying the number of clusters $K$, the standard deviation $\sigma$ for the Gaussian randomization of margins and the hypersphere radius $s$ for CelebA and Waterbirds (with ResNet-18 as the feature encoder) in Fig. 4. We find that performance remains consistent for $K \geq 4$ for both datasets. Similarly, similar scores are seen with $\sigma > 0.1$. The best values for $s$ are found to be 8 and 12. We also vary the weight decay $\lambda$ in the bias amplification stage and show its effect on the final model performance. While for CelebA, the effect of changing $\lambda$ is minimal, Waterbirds can be seen to be more sensitive towards higher values.

## A.4 Performance on Other Datasets and Architecture

Here we describe two real-world datasets: UTKFace [74] and BAR [20] and compare our model against the ERM model and the Contrastive Adapter [50].

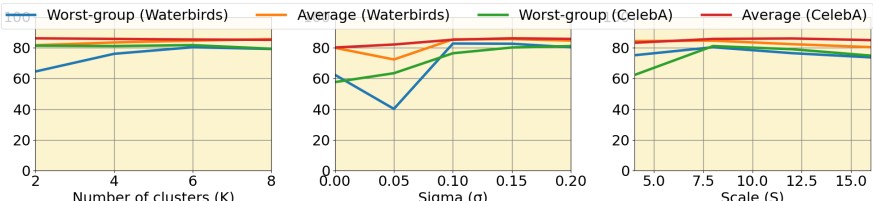

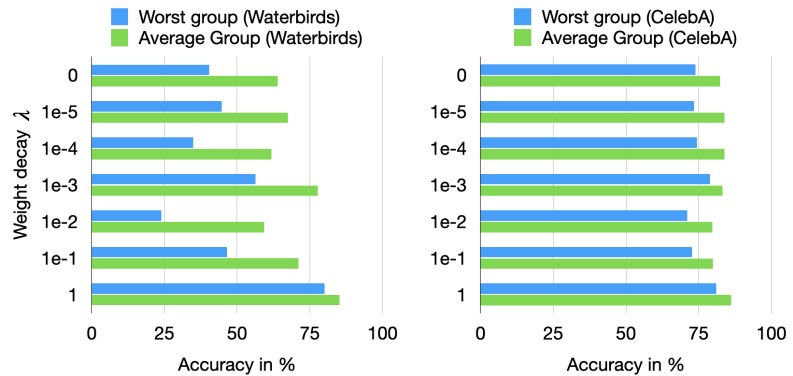

Figure 4: **Hyperparameter analysis**. We vary the number of clusters $K$, the $\sigma$ for the Gaussian randomization of margins and the hypersphere radius $s$ and show model performance for Waterbirds and CelebA.

Figure 5: **Effect of Weight Decay on Downstream Model Performance**. We vary the weight decay $\lambda$ and show the final model performance for Waterbirds and CelebA.

### A.4.1 UTKFace

In the main paper, we evaluate on Waterbirds, CelebA (for Blond Hair classification) and CMNIST-0.9, CMNIST-0.995. Here, we evaluate on UTKFace [74], which is a dataset of faces, with gender, race and age annotations. Specifically, we predict race, with gender as the bias attribute. The dataset is stored in 3 parts with 10137, 10719 and 3252 images respectively; we use the first part as the training set, second part as test, and third part as the validation set. Like all other experiments, we manipulate the validation set to contain group-balanced data. There are 5 races - White, Black, Asian, Indian and Other, and two perceived genders - females and males. Out of the 10 resultant groups, we report the worst group's accuracy as well as the average group score. We observe that our method far outperforms the ERM scores (as well as the competing Contrastive Adapter) for the ResNet-18 image encoder, thus further demonstrating the effectiveness of our method. The results are summarized in Table 12.

### A.4.2 BAR

BAR (Biased Action Recognition) [20] is an image dataset with actions as the class labels and places as the spurious attribute. Note that the test set only consists of bias-conflicting samples, hence, we

Table 9: Hyperparameters for the Bias-amplification stage for Waterbirds and CelebA for all the pretrained models shown in the main paper.

|  | Waterbirds | | | | | CelebA | | | | |
|---|---|---|---|---|---|---|---|---|---|---|
|  | LR | $\lambda$ | BS | Ep | Opt | LR | $\lambda$ | BS | Ep | Opt |
| R18 | 0.0001 | 1 | 256 | 100 | Adam | 0.01 | 1 | 512 | 50 | SGD |
| ViT | 0.05 | 1 | 256 | 100 | Adam | 0.0001 | 1 | 512 | 50 | SGD |
| CLIP | 0.01 | 0.1 | 256 | 100 | Adam | 0.0001 | 0.1 | 512 | 50 | SGD |

Table 10: Hyperparameters for the Bias-amplification stage for CMNIST-0.9 and CMNIST-0.995 for all the pretrained models shown in the main paper.

| | CMNIST-0.9 | | | | | CMNIST-0.995 | | | | |
|---|---|---|---|---|---|---|---|---|---|---|
| | LR | $\lambda$ | BS | Ep | Opt | LR | $\lambda$ | BS | Ep | Opt |
| R18 | 0.01 | 0.05 | 256 | 100 | Adam | 0.0001 | 0.1 | 256 | 100 | Adam |
| ViT | 0.01 | 0.01 | 256 | 100 | Adam | 0.0001 | 0.1 | 256 | 100 | Adam |
| CLIP | 0.01 | 0.01 | 128 | 100 | Adam | 0.0001 | 0.01 | 256 | 100 | Adam |

Table 11: Hyperparameters for Mitigation stage of our approach for Waterbirds, CelebA, CMNIST-0.9 and CMNIST-0.995.

| Datasets | Pretrained Encoder | LR | $\lambda$ | BS | Eps | Opt | K | s | $\sigma$ |
|---|---|---|---|---|---|---|---|---|---|
| | R18 | 0.01 | 0.01 | 64 | 100 | Adam | 6 | 8 | 0.2 |
| Waterbirds | ViT | 0.01 | 0.01 | 128 | 100 | Adam | 4 | 8 | 0.25 |
| | CLIP | 0.01 | 0.01 | 256 | 100 | Adam | 4 | 8 | 0.2 |
| | R18 | 0.0001 | 0.0001 | 128 | 100 | SGD | 4 | 8 | 0.2 |
| CelebA | ViT | 0.01 | 0 | 128 | 100 | SGD | 2 | 8 | 0.15 |
| | CLIP | 0.1 | $1e-5$ | 128 | 100 | SGD | 2 | 8 | 0.2 |
| | R18 | 0.01 | 0 | 128 | 100 | Adam | 30 | 12 | 0.15 |
| CMNIST-0.9 | ViT | 0.0001 | $1e-6$ | 256 | 100 | Adam | 20 | 4 | 0.15 |
| | CLIP | 0.0001 | $1e-6$ | 256 | 100 | Adam | 20 | 8 | 0.2 |
| | R18 | 0.01 | 0 | 256 | 100 | Adam | 20 | 8 | 0 |
| CMNIST-0.995 | ViT | 0.005 | $1e-4$ | 128 | 100 | Adam | 40 | 8 | 0.05 |
| | CLIP | 0.001 | 0 | 128 | 100 | Adam | 40 | 12 | 0.1 |

Table 12: Performance of our method on the UTKFace dataset and the BAR dataset. In all cases, we find our proposed approach outperforming the worst group accuracies of the ERM method and Contrastive Adapter. The underlying pretrained encoder is the ResNet-18 model.

| Method | UTKFace | | BAR |
|---|---|---|---|
| | Worst Group | Avg Group | Test |
| ERM | 13.6 | 40.5 | 63.15 |
| Contrastive Adapter | 8.36 | 31.46 | 62.54 |
| Our method | **32.26** | **40.65** | **65.96** |

report its test accuracies following previous work [38]. As with UTKFace, we find that our method outperforms both the ERM method and the Contrastive Adapter (see Table 12).

### A.4.3 ViT-H 14 as the Pretrained Encoder

In a previous work [61], it is shown that spurious correlations can be mitigated simply by using a stronger pretrained encoder. Further, the authors show that the strength of mitigation depends on the underlying pretraining data. They observe that using a ViT-H 14 encoder pretrained on the SWAG dataset [62] followed by end-to-end ImageNet [63] finetuning can achieve state-of-the-art results on the Waterbirds dataset. Likewise, to test whether the same holds for other datasets too, we use this same encoder and train for CelebA and CMNIST-0.995 using the ERM method. We observe that while indeed the worst group score for Waterbirds is considerably high, for CelebA and CMNIST-0.995, the performance is still weak. This leads us to the important observation that bias mitigation strategies are indeed required for all pretrained encoders, as even though they might perform well for some biased datasets, they may not be free of all biases that can affect model predictions. We compare the ERM scores with that of our margin loss approach and Contrastive Adapter, and find that in Waterbirds, though the ERM scores outperform the mitigation methods marginally, for the other two datasets, the mitigation scores considerably improve. Upon comparing our method with Contrastive Adapter, we find that the former outperforms the latter in CMNIST-0.995 and Waterbirds. The results are shown in Table 13.

Table 13: Performance on ViT-H 14 (pretrained on SWAG and finetuned on ImageNet). In this table we show the performance of ERM, Contrastive Adapter and our margin loss-based method for Waterbirds, CelebA and CMNIST-0.995. We observe that while the ERM scores are high for Waterbirds, the same does not hold for CelebA and CMNIST-0.995 – showing that a single feature encoder, however powerful, may not mitigate all biases. Thus we advocate strongly for bias mitigation strategies that can debias models in presence of untrainable feature encoders.

| Method | Waterbirds | | CelebA | | CMNIST-0.995 | | |
|---|---|---|---|---|---|---|---|
| | Worst Group | Avg Group | Worst Group | Avg Group | Bi-Co | Bi-Al | Avg |
| ERM | **91.1** | **95.99** | 6.71 | 61.32 | 68.41 | **100** | 71.57 |
| Contrastive Adapter | 86.12 | 93.17 | **82.23** | **85.74** | 78.09 | 94.97 | 86.53 |
| Our method | 89.17 | 95.26 | 62.93 | 68.61 | **85.51** | 95.29 | **90.4** |

Table 14: **Ablations of our method**: Here we further show the roles of the different components of our margin loss-based approach using ResNet-18 as the pretrained backbone.

| Model Component | Waterbirds | | CelebA | | CMNIST-0.9 | | |
|---|---|---|---|---|---|---|---|
| | Worst | Average | Worst | Average | Bi-Co | Bi-Al | Average |
| Our method | 80.29 | 84.56 | 81.61 | 86.04 | 81.91 | 92.75 | 87.33 |
| Randomization (margin = 0.0) | 40.34 | 76.26 | 25.56 | 75.45 | 55.65 | 99.68 | 77.67 |
| Randomization (margin = 1.0) | 40.34 | 76.72 | 30.00 | 76.78 | 56.48 | 99.69 | 78.09 |
| Randomization (margin = 0.5) | 43.61 | 76.78 | 29.44 | 76.20 | 55.61 | 99.68 | 77.65 |

## A.5 Ablations

In this subsection, we further discuss the importance of the adaptive margin components of our approach and the margin loss strategy. Specifically, we present the results when the randomization is applied to constant margin values. As with Table 6 in the main paper, all evaluations are performed on Waterbirds, CelebA (Blond Hair classification) and CMNIST-0.9 with ResNet-18 as the backbone. The results are shown in Table 14.

**Observations**. Here we show the effects of randomizing a constant margin value to further understand the role of the *adaptive* margins. We perform this experiment on 3 constant margins – 0.0, 1.0 and 0.5. We observe that for all three datasets, both worst and average accuracies decrease drastically. This shows the effectiveness of the *adaptive* margins, i.e., randomizing the margins do not help unless their underlying values represent the frequency of a training sample's class label in its assigned cluster.

We next show further ablations of our method. As described in the main paper, in the bias-amplification stage, we set the weight decay $\lambda$ to a high value (weight decay is the weight used for $L2$ regularization) to ensure amplification of the bias in the model. Instead of the $L2$ loss, we investigate what happens when we instead use $L1$ and $L1 + L2$ regularization during this stage. The performances for all these different regularizations appear similar. We also show what happens when we do not use any of these regularizations – the scores drop drastically for Waterbirds. Moreover, in many of the previous works like JTT [19], misclassifications in the ERM stage are used to identify the biases. When we use the same in our method to calculate the margins (instead of clustering), we find that clustering works better than using misclassifications for bias-identification. Further, we find that using the Generalized Cross-Entropy loss [20] with $L2$ regularization can have similar effects as that of our original approach, though the latter outperforms the former for both the datasets. We also investigate what happens when we replace the ArcFace loss [6] with another margin loss called CosFace [5]. Here we only change the loss formulation, but calculate the margins in the same way as that our method as described in the main paper. We call this version of our method CM + CosFace, and find that it is indeed effective, however the original margin loss still outperforms the former on both Waterbirds and CelebA. The results are summarized in Table 15.

## A.6 Comparison with Other Methods

### A.6.1 Last Layer Retraining

In DFR [54], Kirichenko et al. show that simply retraining the last layer of a biased model can achieve considerably unbiased predictions. However, they utilize a group-balanced reweighting

Table 15: In this table we summarize the roles of the different regularization techniques as well as misclassifications as an alternate for clustering on the ResNet-18 pretrained encoder. We find that using L1 regularization can be a good (even better) proxy for weight decay. We also show what happens when we use the GCE loss [20] along with high weight decay for the bias amplification stage. Finally, we evaluate the changes in performance when we use an alternate margin loss to mitigate the bias in the network.

| Method | Waterbirds | | CelebA | |
|---|---|---|---|---|
| | Worst Group | Avg Group | Worst Group | Avg Group |
| L1 Regularization | 81.80 | 85.21 | 81.79 | 85.50 |
| (L1+L2) Regularization | 78.55 | 85.66 | 79.98 | 85.47 |
| No Regularization | 39.88 | 77.04 | 78.33 | 80.71 |
| Misclassification | 75.86 | 82.76 | 75.56 | 85.23 |
| GCE + L2 Regularization | 78.35 | 83.43 | 78.33 | 83.28 |
| CM + CosFace | 77.07 | 82.55 | 75.56 | 85.42 |
| Our method | 80.29 | 84.56 | 81.61 | 86.03 |

Table 16: In this table we show the performance comparison of DFR with the proposed approach for the ResNet-18 backbone, for Waterbirds, CelebA and CMNIST-0.9

| Method | Waterbirds | | CelebA | | CMNIST-0.9 | | |
|---|---|---|---|---|---|---|---|
| | Worst Group | Avg Group | Worst Group | Avg Group | Bi-Co | Bi-Al | Avg |
| Our method | **80.29** | **84.56** | **81.61** | **86.04** | 81.91 | **92.75** | 87.33 |
| DFR | 74.68 | 81.29 | 73.33 | 80.78 | **91.39** | 90.84 | **91.16** |

dataset (specifically from the validation set) for this retraining. The difference of this setting with ours is that ours does not assume a feature encoder which is already trained on the biased dataset; rather we assume a blackbox feature encoder which does not interact with the downstream training data. We summarize the results in Table 16. While we see that the performance of Waterbirds and CelebA are comparable to our method, DFR has superior performance in CMNIST-0.9. However, it is to be remembered here that DFR not only uses part of the validation data for training, but also this excess data is group-annotated. This gives additional advantage to DFR, leading to improved scores in some datasets.

### A.6.2 Contrastive Adapter

Table 17: Difference in training time (**in mins**) for Contrastive Adapter and the proposed approach for different datasets trained on the ResNet-18 backbone. We find that while our method is trained in considerably less amount of time, Contrastive Adapter requires enormous time, rendering it inefficient for time-constrained settings.

| Dataset | Contrastive Adapter | Our method | Gap |
|---|---|---|---|
| Waterbirds | 34 | 1.05 | **32.95** |
| CelebA | 105 | 13.39 | **91.61** |
| CMNIST | 213 | 2.05 | **210.95** |

While the problem statement described by Zhang et al. [50] is similar to ours, there are subtle differences. E.g., Contrastive Adapter is meant to boost the group robustness of foundation models. It uses the zero-shot capabilities of the existing foundation models (as a result, utilizes their text modalities) to find positive and negative samples with respect to an anchor sample and apply a contrastive loss for bias mitigation purposes. On the other hand, our aim is more generic: to detect and mitigate biases in case the feature encoder is untrainable (only accessible through API calls) and does not rely on any other modality. That is why, when we compare our method against other methods, we freeze the feature encoder for all of them and investigate how these methods perform when no backpropagation is allowed into the frozen backbone. Moreover, since ResNet-18 and ViT-Base do not have the text encoder, we cluster the backbone features as a proxy for the zero-shot

accuracies of Contrastive Adapter using the KMeans algorithm (suggested by the authors in their code submission). Since KMeans is an unsupervised algorithm, the cluster label might not always correspond to the correct class index. Simple heuristics are employed to properly label each obtained cluster. In the binary classification setting (WaterBirds and CelebA), the cluster that has a higher number of correct predictions is assigned the label 0 (since WaterBirds has a higher number of landbirds (class 0) and CelebA has a higher number of Non-blonde samples (class 0)). This heuristic fails when the number of classes exceeds 2 (e.g. ColorMNIST), and in such cases, we use consensus to handle the prediction. For each cluster, we count the number of predictions for each class (digit), and the cluster label is the class with the largest number. This associates the most probable label with each cluster in the KMeans model – i.e., we take the ground truth class (digit) of each sample, find the majority class per cluster and assign that digit as the predicted class.

**Impact of number of positives and negatives.** Contrastive Adapter requires the creation of a similarity matrix, which consists of positive and negative samples for every sample in the dataset. Typically, a reasonable number of positive and negative samples are necessary for the method (contrastive learning) to function properly. The caveat is that when dealing with multi-class classification problems, gathering a sufficient number of negative samples takes a toll on the system memory, and a server of 128GiB can support only about 128 positives and negatives before overflowing. In addition to this, even if a sufficiently large server is procured to be able to support a higher number of samples, the training time grows significantly, which is a further drawback on the total runtime as well.

Table 18: **Effect of full finetuning.** Upon finetuning the encoders fully, our method achieves decent performance compared to the existing methods across two pretrained encoders. The numbers of all competing methods are taken from previous works for ResNet-18 [45] and ResNet-50 [19].

| Method | Res-18 | | Res-50 | |
|---|---|---|---|---|
| | Worst | Avg | Worst | Avg |
| ERM | 62.39 | 84.63 | 72.6 | _97.3_ |
| LfF | 68.02 | 85.48 | 75.2 | **97.5** |
| BPA | _71.39_ | **87.05** | - | - |
| JTT | - | - | _86.0_ | 93.6 |
| Our method | **80.82** | _85.91_ | **88.82** | 90.62 |

We show the differences in training times of the proposed method and Contrastive Adapter in Table 17. It is to be noted that we perform all our experiments on a single Nvidia-RTX A5000 GPU. In a time-constrained setting, this renders Contrastive Adapter inefficient, whereas our method, even with considerable less training time, outperforms Contrastive Adapter in most cases.

## A.7 Full finetuning

In this subsection, we discuss the effect of full finetuning on the proposed loss, i.e., when the loss is allowed to backpropagate into the pretrained encoder. We compare our method with LfF [20], BPA [45] and JTT [19]. We report the results for LfF and BPA on the ResNet-18 encoder as per the results shown by Seo et al [45], whereas those for JTT (along with LfF) are with respect to the ResNet-50 model [19]. From Table 18, we note that for ResNet-18, our method outperforms BPA and LfF. For ResNet-50, our method surpasses that of JTT by $3\%$ in the worst group accuracy. In comparison, for our proposed setting, our method outperforms JTT by a huge margin ($\mathbf{30.45\%}$)! This study shows how our method performs satisfactorily in the setting of full finetuning, in addition to decent and consistent performance on the problem setting proposed in the paper.

## A.8 Social Impact

Since our work aims to create unbiased models in a unique problem setting, we believe that it promotes a positive social impact. Pretrained models are not free of biases [14], and our method is an attempt to generate unbiased predictions without having to finetune these pretrained encoders – we believe that this is a step towards ensuring that future deep learning models are free of the prejudices already present in the society towards marginalized groups, and are fair to all.

