# OpenReview forum: "Mitigating Biases in Blackbox Feature Extractors for Image Classification Tasks"
_NeurIPS.cc/2024/Conference — NeurIPS 2024 poster_

### Official Review · Reviewer_oZ3K · 2024-07-08

**Soundness:** 3
**Presentation:** 3
**Contribution:** 3
**Rating:** 6
**Confidence:** 3

**Summary:**

The paper tries to address the critical issue of biases in blackbox feature extractors used for image classification tasks. These biases can impact the performance of models when adapted for downstream tasks. The authors investigate existing debiasing techniques and propose a novel method using a clustering-based adaptive margin loss, which does not require prior knowledge of bias attributes. Their experiments demonstrate the effectiveness of this approach across multiple benchmarks, highlighting its practical applicability in scenarios where feature extractor weights are not accessible.

**Strengths:**

(1) The paper's shows originality and practical relevance. It tackles a challenging problem with a novel approach that is both effective and efficient. The clustering-based adaptive margin loss is a creative solution that demonstrates substantial improvements over existing methods. The thorough experimental validation across multiple benchmarks further strengthens the paper's contributions.

(2) The paper is well-written and clearly presented. The introduction effectively sets the stage for the problem being addressed, and the methodology is detailed comprehensively.

**Weaknesses:**

(1) The paper could benefit from a deeper theoretical analysis of why the proposed method works, which would enhance its scientific rigor.

**Questions:**

(1) How does the clustering-based adaptive margin losses compare to other adaptive loss functions in terms of computational complexity?
(2) Could the proposed method be applied to other types of pre-trained models beyond image classification tasks? If so, what adjustments would be necessary?
(3) How sensitive is the performance of the proposed method to the choice of clustering algorithm and the number of clusters?

**Limitations:**

(1) A more detailed discussion on the scalability of their approach to extremely large datasets and models would be beneficial.
(2) While the method shows effectiveness in mitigating biases, exploring its impact on model interpretability could provide valuable insights.

---

> ### Author Rebuttal · Authors · 2024-08-06
>
> We thank the reviewer for the thoughtful comments. We address the concerns below:
>
> **Deeper analysis of the approach.**
> We refer the reviewer to the ArcFace paper [6] for details on the margin loss. Here, we show how the adaptive nature of the margin loss aids in the learning of the bias-conflicting samples. As shown in [6], the margin loss increases the angle between a sample's MLP feature ($\hat{f}$) and its ground truth weight vector. The degree of increase depends on which cluster the sample belongs to, and the inverse frequency of the ground truth class in that cluster. Let $x$ be an image in the bias-conflicting group, $m$ be its corresponding margin penalty (eq 2 and L248 in the main paper), and $\theta$ be the angle between the weight vector corresponding to the ground truth class and the feature $\hat{f}$. As $x$ belongs to a bias-conflicting group, the penalty $m$ is high (assuming that the clusters obtained from the features are good approximations of the different groups in the training set), which means that the angle between the feature vector and its weight vector increases considerably in the softmax term inside the margin loss in eq 4 (main paper) (recall that $\cos(\theta+m) < \cos(\theta)$ for $m>0$). Thus the loss value rises for the bias-conflicting samples (owing to the logarithm and the negative sign in eq 4). For the bias-aligned samples, as $m$ is lesser (as it belongs to the majority class in the cluster), the degree of increase in the loss is lower than those for the bias-conflicting samples. This is how the adaptive margins help mitigate biases.
>
> For further evidence on the usefulness of the adaptive margin loss, we plot the training loss for each group (summed across all epochs and normalized by group size) in Waterbirds and CelebA between an ERM model and our method in Figure 2 in the rebuttal pdf. While both models have higher losses for the minority groups (for majority groups, the loss values are negligible and similar), the proposed method's minority loss values are much higher than that of the ERM model.
>
> **Computational Complexity of competing methods**
> See the below table for the analysis of computational complexity.
>
> |    Time    |  ERM |  BPA |  LfF |    CoAda    | Ours |
> |:----------:|:----:|:----:|:----:|:-----------:|:----:|
> | WaterBirds |  22s |  48s |  44s |     615s    |  65s |
> |   CelebA   | 114s | 400s | 832s | 127 mins | 803s |
>
> |     RAM    |  ERM  |  BPA  |   LfF  | CoAda |  Ours |
> |:----------:|:-----:|:-----:|:------:|:-----:|:-----:|
> | WaterBirds | 1.28G | 1.34G | 0.890G |  7.5G |  1.3G |
> |   CelebA   | 1.71G | 3.87G |  1.28G |  112G | 1.82G |
>
> |    VRAM    | ERM |  BPA | LfF | CoAda | Ours |
> |:----------:|:---:|:----:|:---:|:-----:|:----:|
> | WaterBirds | 25M |  28M | 20M |  44M  |  20M |
> |   CelebA   | 20M | 752M | 20M |  34M  |  20M |
>
> **Application to other pretrained models**
> In the main paper, we show the performance of our model on self-supervised pretrained model like CLIP, and demonstrate the effectiveness of our method in Table 4. Our method can further be applied to any classification based task if a pretrained feature encoder is present. For example, if one wants to run it for a text-based classification task, they only need the features from a pretrained model (e.g., BERT). No other adjustment is necessary. It is to be noted that the method only relies on the angle between the feature vector and the ground truth class' weight vector. As an application, we run our method on the CivilComments dataset [a] using the BERT model pretrained on BookCorpus and Wikipedia datasets (https://huggingface.co/google-bert/bert-base-uncased), and report the results below. We find that both Contrastive Adapter and our method far outperform the ERM scores.
>
> |       | Average | Worst |
> |-------|---------|-------|
> | ERM   |   58.14 |  13.5 |
> | CoAda |   78.19 | 67.96 |
> | Ours  |   76.43 | 68.06 |
>
> **Clarification for the clustering task** Kindly see the general response section.
>
> **Scalability of approach to extremely large datasets and models**
> Since the method involves clustering the features, if the dataset is extremely large, one can randomly sample a small percentage of it to do the clustering. For example, on clustering only 10% randomly sampled images from celeba, we find that the worst group accuracy is $81.11$%, whereas the average group accuracy is $85.6$%. The scores become $80.55$% and $85.6$% respectively when the clustering is performed only on 1% of the images (Note that scores reported in the paper are $81.61$% and $86.04$% respectively). On the other hand, for extremely large models, the requirement is to be able to load the model into a GPU memory. Our method adds negligible overhead owing to the addition of only an adapter and a classifier layer. We will add these points to the limitations section in the camera ready version.
>
> **Exploring the method impact on model interpretability**
> We thank the reviewer for this valuable suggestion. We intend to investigate this as a future work.
>
>
>     [a] Borkan, Daniel, et al. "Nuanced metrics for measuring unintended bias with real data for text classification." Companion proceedings of the 2019 world wide web conference. 2019.

---

> > ### Author Response · Authors · 2024-08-14
> > **Official Comment by the Authors**
> >
> > We once again appreciate the thoughtful feedback provided by the reviewer. We hope our responses have adequately addressed the concerns of the reviewer.

---

> ### Comment · Area_Chair_msYo · 2024-08-08
> **Please read the rebuttal to check if the authors addressed your concerns**
>
> Dear Reviewer oZ3K,
>
> Can you have a look at the rebuttal and see if your concerns have been addressed?
>
> Best regards
> Your AC.

---

### Official Review · Reviewer_wKPv · 2024-07-11

**Soundness:** 3
**Presentation:** 2
**Contribution:** 3
**Rating:** 5
**Confidence:** 3

**Summary:**

In this paper, the authors propose a simple method with a clustering-based adaptive margin loss for debiasing blackbox pretrained models. Whereas prior works have explored settings where pretrained models are tunable, the authors instead explore a more constrained and realistic setting, where a black-box network is frozen and only a classifier is trained on the fine-tuning dataset. The proposed approach involves training an adapter to amplify biases in the frozen encoder; then, a mitigation procedure is executed using a novel cluster-based margin loss. The proposed approach appears to outperform state-of-the-art methods across three datasets.

**Strengths:**

1. The paper introduces a creative approach for addressing a high-impact problem that has been previously under-researched. As large-scale pretrained encoders become more commonplace, it is important to create debiasing methods that can operate in settings where the model weights are frozen.
2. The proposed approach appears to demonstrate performance improvements over existing approaches while also demonstrating high efficiency.

**Weaknesses:**

1. **Applicability**: One important weakness of this approach is that it is difficult to know when it can be used. The authors claim that their "method is specifically targeted towards cases where the bias in the downstream dataset is already encoded in the pretrained model". However, this is difficult (and in some cases, impossible) to know a priori.
    - In their evaluations in Table 1, the authors assume that spurious attribute labels are available in order to determine whether a pretrained model encodes the same bias as the downstream dataset. However, these subgroup labels are not likely to be available in real-world settings (which the authors claim as well later in the paper), making it difficult to know when the authors' approach is useful.
    - The authors also make a note about how their adapter-based bias amplification procedure will consistently work because they assume that the bias in the downstream dataset aligns with that of the pretrained model. As stated above, this is a strong assumption that is difficult to verify a priori, especially since the authors assume that bias annotations are not available for their downstream dataset.
    - As a result, I am unsure on the real-world applicability of this approach.
2. Presentation: The organization of this manuscript is somewhat confusing, particularly in section 3.
3. Additional clarification on design decisions: Some critical design decisions are somewhat vague in the main text.
    - How was the value of $\lambda$ selected in practice for each dataset? The paper states that $\lambda$ was selected as a high value while ensuring that the training accuracy did not drop drastically. Are there specific thresholds at which the training accuracy drop was viewed as "drastic"?
    - The paper could benefit from additional details on the clustering approach. How was the number of clusters selected? In Table 11, it appears that the number of clusters K selected by the authors varies significantly across encoders and datasets. Is this a hyperparameter that needs to be set by the user? How does performance vary depending on the number of clusters selected? It seems that this is a critical design decision that can affect efficacy of the proposed approach.

**Questions:**

I have listed my questions above in the “weaknesses” section.

**Limitations:**

Yes, the authors have adequately described the limitations of their work.

---

> ### Author Rebuttal · Authors · 2024-08-06
>
> We thank the reviewer for the thoughtful comments. We address the concerns below:
>
> **Clarifications about Table 1:** Table 1 is solely meant for analyzing the nature of the different feature encoders, wherein we can see that different feature encoders may have different levels of awareness of the target labels. If a feature encoder is highly target-aware, even removing all bias-conflicting samples from the training data does not hamper the worst group accuracies to a large extent. On the other hand, if the feature encoder is bias-aware, the worst group accuracies are low even for the original dataset -- these are the scenarios that require explicit bias mitigation. Note that the scores in this table are those of the ERM method, obtained by varying the proportion of the bias-conflicting samples, and is not related to the proposed approach.
>
> **Assumption on alignment of the bias in the pretrained features and downstream dataset:**
> Please see the general response section.
>
> **Clarifications on the presentation in Section 3:**
> We apologize if the presentation of this section was not clear. We explain the main highlights of the section below and will make the presentation clearer in the camera-ready version upon acceptance.
>
> - We begin by formally defining the existence of biased groups in deep learning datasets, explaining how model behavior differs for different groups (section 3.1).
> - As mentioned above, section 3.2 analyzes the nature of different feature encoders, and highlights that the bias-aware feature encoders require explicit mitigation.
> - In section 3.3, we show how existing methods (designed for full finetuning) perform inconsistently for different datasets in the proposed problem setting, while Contrastive Adapter [50] (designed for such frozen encoders) performs decently, but is highly time consuming. This motivates the requirement of dedicated research in the direction of the proposed problem setting.
> - Section 3.4 describes our method, where we first motivate amplifying the bias, and then mitigating it through the clustering-based margin loss approach, after exploring a number of simple baselines involving weighted losses.
>
> We can clarify further if the reviewer still has doubts.
>
> **Clarifications on selection of weight decay value:**
> In the paper, we have shown that high weight decay leads to a reduction in the bias-conflicting group performances in the training set, thereby helping us detect the bias easily. The value of $\lambda$ is chosen using the validation accuracy in the mitigation stage in practice from a range of high values (0.01-1). Details are present in Appendix A.4. When $\lambda$ is sufficiently high, we find that the training accuracy collapses (~50% for Waterbirds and CelebA, <30 % for ColorMNIST-0.995). This fall can be clearly seen in Figure 2c in the main paper for ColorMNIST-0.995 for $\lambda \geq 0.5$. This helps us eliminate a few values of $\lambda$ in our specified range during the amplification stage itself. For Waterbirds and CelebA, the fall is typically seen at $\lambda \geq 3$ for the ResNet-18 backbone. This happens as the model’s learning is impeded with extremely high weight decay, which is not the goal of our approach. Recall that the purpose of this step is to learn the bias-aligned samples well, not the bias-conflicting samples. Hence, in the paper we suggest picking $\lambda$ lower than this threshold. We stick to the search space of (0.01-1) as we find that the scores remain similar for $\lambda \geq 1$ for both datasets (see below).
>
> | $\lambda$ | Worst - Waterbirds | Avg - Waterbirds | Worst- CelebA | Avg - CelebA |
> |:---------:|:------------------:|:----------------:|:-------------:|:------------:|
> |     1     |        80.29       |       84.56      |     81.61     |     86.04    |
> |    1.5    |        79.44       |       84.52      |     81.87     |     85.35    |
> |     2     |        81.46       |       84.59      |     80.51     |     85.21    |
>
>
> **Clarifications on the clustering algorithm:**
> Please see the general response section.

---

> ### Comment · Area_Chair_msYo · 2024-08-08
> **Please read the rebuttal to check if the authors addressed your concerns**
>
> Dear Reviewer wKPv,
>
> Can you have a look at the rebuttal and see if your concerns have been addressed?
>
> Best regards
> Your AC.

---

> > ### Comment · Reviewer_wKPv · 2024-08-13
> > **Response to authors**
> >
> > I thank the authors for their detailed responses, which have addressed most of my concerns. However, my concerns on the applicability of this approach still stand, since it is difficult to know a priori whether the method will be effective. I will maintain my rating.

---

> > > ### Author Response · Authors · 2024-08-13
> > > **Response to the reviewer**
> > >
> > > We thank the reviewer for the response. As the reviewer pointed out in the original review, we agree that identifying if the bias in the pretrained encoder aligns with that in the downstream dataset is highly difficult. This challenge is posed due to the nature of the proposed problem setting, since no finetuning/backpropagation is allowed into the pretrained encoder. In this regard, during our experiments, we find that the features of the ViT-B backbone do not align very strongly with the bias in Waterbirds (we refer the reviewer to the general responses section for more details). As a consequence, we find that most competing methods are unable to outperform the ERM method. Our method is able to surpass the ERM scores by a significant margin, though the scores are not higher than that of the ResNet-18 backbone for the same dataset. This finding highlights the difficulty of this problem setting, and we firmly believe that answering whether it is possible to mitigate biases when they are not encoded in the pretrained model is a very vital research direction.

---

### Official Review · Reviewer_qb1H · 2024-07-13

**Soundness:** 4
**Presentation:** 4
**Contribution:** 4
**Rating:** 8
**Confidence:** 4

**Summary:**

This work explores a problem setting where one wants to train a classifier on top of a large and frozen pre-trained model, while avoiding bias and improving fairness. It proposes a computationally efficient methodology to accomplish this objective, centered around a novel loss function, the Adaptive Margin Loss.

**Strengths:**

-The manuscript is very clear and well-written.
-Multiple experiments were performed.
-The proposed methodology seems computationally efficient and effective at avoiding bias.
-Results are promising, and the objective of adapting large pre-trained models in an unbiased manner will probably be increasingly more important in the future, with the rise of such pre-trained models.

**Weaknesses:**

My comments here are minor suggestions, as I have no major concerns.

1- Figure 2: this analysis is very interesting. It would be nice to start the weight decay as 0, not 5e-2 in the plots.

2- A question I have is, how many unbiased samples do we need for this methodology (and the alternative ones) to work well?

Table 8 shows some results that help us answer this question, as we can see how performances vary from CMNIST-0.9 to CMNIST-0.995. What would happen for CMNIST-0.999? Where is the limit where the debiasing methodologies fail?

In summary, it would be great to expand these experiments, varying the proportion of the majority group in all datasets (not only CMNIST, but also CelebA and Waterbirds), until the point where all methods fail. This experiment would show the reliance of each methodology on the number of samples in the minority group.

3- In the tables, we have standard deviations only for the proposed methodology. They should also be reported for the competing methods as well. Moreover, although authors say that “All experiments have been done over 3 seeds”, multiple tables show no standard deviation.

**Questions:**

My main question is why we do not see standard deviations in most tables, if all experiments have been done over 3 seeds?

**Limitations:**

Authors have correctly stated the method's limitations.

---

> ### Author Rebuttal · Authors · 2024-08-06
>
> We thank the reviewer for the positive comments and feedback and address the specific concerns below.
>
> **Effects of starting from weight decay $\lambda=0$ in Figure 2 of main paper**: We have shown the effects starting from weight decay = 0 on the different group accuracies for Waterbirds, CelebA and ColorMNIST-0.995 in Figure 1 in the rebuttal pdf, where we see that the lower the weight decay, the higher the training accuracy, because of overfitting to all the groups.
>
> **Performance of different models for varying degree of bias-conflicting samples**
> This is a great suggestion, and likewise we train our method on ColorMNIST-0.999 (Table 1, rebuttal pdf), and two versions of Waterbirds and CelebA. For Waterbirds, we show performance of different methods with a) no bias-conflicting samples, b) 50% bias-conflicting samples, where the bias-conflicting samples constitute two groups: waterbirds on land and landbirds on water (Table 2, rebuttal pdf). We show the scores for similar versions of CelebA, where the bias-conflicting samples constitute the blond males (Table 3, rebuttal pdf). As predicted by the reviewer, with a decrease in percentage of bias-conflicting samples, the scores of all methods drop. In Waterbirds, for the case of no bias-conflicting samples, the drop in performance is substantial, as is for ColorMNIST-0.999 (see Table 1, rebuttal pdf). For all other cases, we find Co-Ada and our method to perform decently. It is to be noted that our method outperforms all others in all the explored cases.
>
> **Standard Deviation for competing methods and other tables**
> We will rectify this in the camera-ready version, upon acceptance. We apologize for not showing the standard deviation values in all cases to ensure ease of readability. We show the values for some of the competing methods below (for ResNet-18 backbone).
>
> | Worst Group Accuracy     | ERM               | DebiAN            | BPA               | LfF               | CoAda             | Ours              |
> |:------------:|-------------------|-------------------|-------------------|-------------------|-------------------|-------------------|
> | WaterBirds |   $38.9^{\pm1.4}$ | $58.94^{\pm0.97}$ |  $58.7^{\pm1.67}$ | $66.09^{\pm0.62}$ | $67.57^{\pm1.29}$ | $80.29^{\pm2.5}$  |
> | CelebA     | $27.20^{\pm0.89}$ | $26.10^{\pm1.12}$ | $66.71^{\pm0.44}$ | $13.26^{\pm0.67}$ | $78.37^{\pm1.36}$ | $81.61^{\pm1.02}$ |
>
> | Average Group Accuracy    | ERM               | DebiAN            | BPA               | LfF               | CoAda             | Ours              |
> |:------------:|-------------------|-------------------|-------------------|-------------------|-------------------|-------------------|
> | WaterBirds | $76.22^{\pm1.04}$ | $80.47^{\pm0.97}$ | $80.83^{\pm1.12}$ | $81.39^{\pm0.47}$ | $80.10^{\pm0.14}$ | $84.56^{\pm1.2}$  |
> | CelebA     | $75.43^{\pm0.67}$ | $75.41^{\pm0.44}$ | $84.14^{\pm0.22}$ | $69.42^{\pm0.36}$ | $85.79^{\pm0.78}$ | $86.04^{\pm0.26}$ |

---

> > ### Comment · Reviewer_qb1H · 2024-08-14
> >
> > I appreciate the authors responses, and all my concerns were addressed very well. I am increasing my score to 8.

---

> > > ### Author Response · Authors · 2024-08-14
> > > **Thank you for your response**
> > >
> > > We thank the reviewer for appreciating our rebuttal and updating the score.

---

> ### Comment · Area_Chair_msYo · 2024-08-08
> **Please read the rebuttal to check if the authors addressed your concerns**
>
> Dear Reviewer qb1H,
>
> Can you have a look at the rebuttal and see if your concerns have been addressed?
>
> Best regards
> Your AC.

---

### Official Review · Reviewer_pwPo · 2024-07-13

**Soundness:** 2
**Presentation:** 2
**Contribution:** 3
**Rating:** 6
**Confidence:** 2

**Summary:**

The papers address the debiasing problem using pretrained but frozen feature extractors on downstream applications. Then, they propose a clustering-based method that relies on bias-amplified training through cross-entropy loss. After training, they cluster the biased features and mitigate the biases using the resultant clusters. The experimental protocol considers CIFAR, Waterbirds, and CelebA as evaluation datasets. Also, several ablations exist about distinct loss functions and comparison with literature methods under two scenarios.

**Strengths:**

The authors build upon debiasing literature, emphasizing using a frozen feature extractor. This approach requires that debiasing interventions are applied post-feature extraction, a strategy that forms the core of their methodology. Their experimental design is iterative, with each step informed by empirical results. Despite its simplicity, the authors' contributions are clear and understandable.
The experimental protocol is comprehensive, utilizing diverse datasets, including CIFAR, Waterbirds, CelebA, BAR, and UTKFace, for evaluation. This selection allows for a broad assessment of the proposed method's effectiveness across various contexts. The authors also conduct extensive ablation studies on different loss functions, components of their proposed method, and comparisons with existing literature methods. These studies demonstrate the superior performance of the new ideas in multiple scenarios. Additionally, the authors provide extra details on reproducing the main results in the appendix.

**Weaknesses:**

In my opinion, the main weakness lies in assuming that the bias aligns with the feature encoder. There are many cases where the practitioner does not know the model's or the dataset's biases. Also, the paper lacks instructions on how to identify scenarios that fit the previous assumption.

**Questions:**

- About Table 8, Is there any explanation about ResNet-18 achieving best performances than a Vit-B on waterbirds? I Would expect that ViT achieving higher performances in this dataset.

**Limitations:**

Despite the primary focus being on debiasing, one step of the proposed method involves amplifying existing biases. It introduces a significant risk: if the following step, designed to compensate for this amplification, fails to function as intended (which is a possibility in specific scenarios), the method may ultimately exacerbate the very biases it aims to mitigate. This could potentially allow malicious actors to exploit the technique, leading to harmful outcomes.

---

> ### Author Rebuttal · Authors · 2024-08-06
>
> We thank the reviewer for recognizing our contributions and appreciating the comprehensive experiments. We address the concerns below:
>
> **Assumption on alignment of bias of the downstream dataset with that in the pretrained model**. Please see the general responses.
>
> **Table 8: Scores of ViT-B on Waterbirds**: Please see the general responses.
>
> **Limitation on exacerbation of biases**. We thank the reviewer for pointing out this limitation, where after bias-amplification, if the debiasing module fails, the method may end up exacerbating the bias instead of mitigating them. We agree that this is a limitation in general for most of the existing debiasing methods, and attackers can leverage this for all methods that involve a bias amplification stage. We will mention this in the camera ready version.

---

> ### Comment · Area_Chair_msYo · 2024-08-08
> **Please read the rebuttal to check if the authors addressed your concerns**
>
> Dear Reviewer pwPo,
>
> Can you have a look at the rebuttal and see if your concerns have been addressed?
>
> Best regards
> Your AC.

---

> ### Comment · Reviewer_pwPo · 2024-08-10
>
> Thank you for your response.
>
> The authors have adequately addressed my primary concerns, and I have no further questions. I will maintain my previous rating, but conditioned to 'bias exacerbation' revision.

---

> > ### Author Response · Authors · 2024-08-13
> > **Response to the Reviewer**
> >
> > We thank the reviewer for the response. We will definitely revise our manuscript to include the 'bias exacerbation' issue upon acceptance.

---

### Author Rebuttal · Authors · 2024-08-06

We thank all the reviewers for their thoughtful comments, questions, and suggestions. We are pleased that our work has been appreciated and positively rated. The reviewers recognized the significance and under-researched nature of our problem statement (Reviewer *wKPv*), noting its growing importance (Reviewer *qb1H*). They also appreciated the novelty and creativity of our solution (Reviewers *qb1H*, *wKPv*, *oZ3K*), found our experiments to be comprehensive (Reviewer *pwPo*), and considered the paper well-written (Reviewers *oZ3K*, *qb1H*) . We address the common concerns below:

**Assumption on the alignment of the bias in the pretrained model and the downstream task (*pwPo*, *wKPv*)**: While we agree that our method’s efficacy is dependent on this assumption, we would like to highlight that since the feature encoder is frozen and cannot be finetuned based on the downstream dataset, detecting and mitigating biases in such a scenario becomes challenging. As an example, we discuss the case of ViT-B for Waterbirds here (we appreciate Reviewer *pwPo* for raising this question on *Table 8*). We find the ViT-B features to depict the target class more strongly than the bias class. We verify this by clustering the ViT-B features and comparing the Normalized Mututal Information (NMI : https://scikit-learn.org/stable/modules/generated/sklearn.metrics.normalized_mutual_info_score.html) of the cluster labels with the bias (NMI=0.6) and target labels (NMI=0.62) respectively. The worst group accuracy is still low (59%), indicating a weak yet definitive presence of the bias in the model. As a result, all methods suffer, with most methods scoring close to the ERM values, demonstrating this to be a limitation for the compared methods as well. Co-Ada has a worst group score of 63.91%, which is higher than the other competing methods and the ERM. In contrast, our method achieves 74.92%, compared to 80.29% that we obtain from the ResNet-18 backbone. This highlights the difficulties in mitigating biases in different scenarios in the given setup. We find this problem to be interesting, and mention solving this as a future work in the paper.

We believe that detecting if our assumption holds apriori is highly challenging in the absence of the bias labels. We put forward a few suggestions to identify the scenarios that fit this assumption (*pwPo*, *wKPv*):
- Obtain bias annotations for the small validation set. If the worst group accuracy of the validation set does not reduce substantially with increasing weight decay, it indicates that the features have stronger signals of the target class than that of the bias, making it harder to capture the bias.
- If the bias annotations of the validation set cannot be obtained due to privacy concerns, the overall validation accuracy can indicate strength of the bias. For example, the difference between the validation and training accuracy is 25% for an ERM trained method for the Waterbirds dataset on the ResNet-18 backbone. The higher this difference, the more the indication that the model is overfitting to more and more samples. Such overfitting can indicate that the model is learning the bias in the dataset, thus not generalizing on the bias-conflicting samples.


**Details on the Clustering Approach (*wKPv*, *oZ3K*)**:
A common question that has been raised is on the selection of the number of clusters, K. In the paper, we mention K to be a hyperparameter within a fixed set of values (appendix A.4), selected based on the validation accuracy. We demonstrate in the below table that our method is not highly sensitive to the value of K (*wKPv*, *oZ3K*). Below we show the performance variation for *CelebA* over different values of K. The bold values denote the numbers reported in the paper.

| $K$ | Worst | Avg |
|:---:|:--------------:|:------------:|
|  2  |      81.70     |     86.07    |
|  4  |      **81.61**     |     **86.04**    |
|  6  |      81.36     |     85.96    |
|  8  |      81.08     |     85.93    |

For ColorMNIST-0.995, we show the effect of different K's below (the bold values denote the numbers reported in the paper):

| $K$ | Bias-Conflicting | Avg |
|:---:|:--------------:|:------------:|
|  10  |      73.34     |     84.21    |
|  20  |      **72.56**     |     **84.42**    |
|  30  |      72.17     |     83.59    |
|  40  |      72.54     |     83.49    |

Next, we change the clustering algorithm from KMeans to GMM, and show that the underlying algorithm does not alter the outcome of the method significantly.

| Dataset | Worst/Bias-Conflicting (GMM) | Avg (GMM) | Worst/Bias-Conflicting (KMeans) | Avg (KMeans) |
|:---:|:--------------:|:------------:|:--------------:|:-----------:|
|  CelebA  |      81.11     |     85.19    | **81.61** | **86.04** |
|  CMNIST-0.995  |      73.08     |    84.84    | **72.56** | **84.42** |

**Attached pdf**. We have attached a pdf containing a few tables and plots, based on the questions and comments of Reviewers *qb1H* and *oZ3K*.

---

### Author Response · Authors · 2024-08-13
**Thank you to the reviewers**

As we are nearing the end of the discussion phase, we would like to express our gratitude to every reviewer for their feedback and reviews. We are grateful to Reviewers pwPo and wKPv for acknowledging and responding to our rebuttal. We are eagerly waiting for responses from Reviewers qb1H and oZ3K.

---

### Author Response · Authors · 2024-08-14
**Summary of the discussion phase**

We summarize the rebuttal phase here. Overall, all reviewers have appreciated and positively rated our work. They have recognized the importance of our problem setting and the comprehensive experiments performed. During the rebuttal phase, we have attempted to address most of the concerns raised by the reviewers:

* We have discussed in detail on our assumption on the alignment of the bias in the pretrained model and the downstream task (**pwPo**, **wKPv**), and put forward suggestions on identifying a priori if this alignment is present. Our ViT-B experiments show how the extent of alignment affects all compared methods. Therefore, mitigating bias in the downstream dataset, especially when it does not align with the pretrained encoder, is an important area for future research..
* We have shown our method's performance to remain invariant to the number of clusters required by the KMeans algorithm, and similarly, we found the scores to remain similar when we changed the underlying clustering algorithm from KMeans to GMM (**wKPv**, **oZ3K**).
* Based on the feedback of reviewer **qb1H**, we have also shown that increasing the bias in the datasets reduces performance for all methods. Our experiments demonstrate that the proposed approach outperforms competing methods even with increasing bias ratio.
* We have also shown that the problem setting and our solution is applicable to other classification tasks like text, where we find that on the CivilComments dataset, our method far outperforms the ERM performance (Reviewer **oZ3K**).

Overall, we feel that this research direction is crucial, and we hope our paper leads to more discussions in the community regarding the same.

---

### Decision · Program_Chairs · 2024-09-25

**Decision:**

Accept (poster)

**Comment:**

The paper introduces a computationally efficient training of an unbiased classifier using a large and frozen pre-trained model, where a new loss function is introduced: the adaptive margin loss. The paper received 4 reviews, where the average score is 6.25 (Min: 5, Max: 8), with the following strengths: 1) original extension from the debiasing literature, 2) clear contributions, 3) experimental protocol is comprehensive with many datasets, 4) extensive ablation studies, 5) paper is well written, 6) results are convincing, and 7) problem is of high impact. As for the weaknesses, the paper lists the following issues: 1) strong assumption on the alignment of the bias in the pretrained model and downstream task, 2) missing details about clustering, 3) missing uncertainty results from competing approaches, 4) missing details about design decisions, and 5) missing deeper theoretical analysis. Most issues have been successfully addressed in the rebuttal, so I’m recommending the acceptance of the paper.